# TAL effector driven induction of a *SWEET* gene confers susceptibility to bacterial blight of cotton

Kevin L. Cox[1,2,*], Fanhong Meng[1,2,*], Katherine E. Wilkins[3], Fangjun Li[1,2], Ping Wang[1,2], Nicholas J. Booher[3], Sara C.D. Carpenter[3], Li-Qing Chen[4], Hui Zheng[3], Xiquan Gao[2,5,6], Yi Zheng[7], Zhangjun Fei[7], John Z. Yu[8], Thomas Isakeit[1], Terry Wheeler[1,9], Wolf B. Frommer[10,†], Ping He[2,5], Adam J. Bogdanove[3] & Libo Shan[1,2]

Transcription activator-like (TAL) effectors from *Xanthomonas citri* subsp. *malvacearum* (*Xcm*) are essential for bacterial blight of cotton (BBC). Here, by combining transcriptome profiling with TAL effector-binding element (EBE) prediction, we show that *GhSWEET10*, encoding a functional sucrose transporter, is induced by Avrb6, a TAL effector determining *Xcm* pathogenicity. Activation of *GhSWEET10* by designer TAL effectors (dTALEs) restores virulence of *Xcm avrb6* deletion strains, whereas silencing of *GhSWEET10* compromises cotton susceptibility to infections. A BBC-resistant line carrying an unknown recessive *b6* gene bears the same EBE as the susceptible line, but Avrb6-mediated induction of *GhSWEET10* is reduced, suggesting a unique mechanism underlying *b6*-mediated resistance. We show via an extensive survey of *GhSWEET* transcriptional responsiveness to different *Xcm* field isolates that additional *GhSWEETs* may also be involved in BBC. These findings advance our understanding of the disease and resistance in cotton and may facilitate the development cotton with improved resistance to BBC.

[1] Department of Plant Pathology and Microbiology, Texas A&M University, College Station, Texas 77843, USA. [2] Institute for Plant Genomics and Biotechnology, Texas A&M University, College Station, Texas 77843, USA. [3] Plant Pathology and Plant-Microbe Biology Section, School of Integrative Plant Science, Cornell University, Ithaca, New York 14853, USA. [4] Department of Plant Biology, School of Integrative Biology, University of Illinois at Urbana-Champaign, Champaign, Illinois 61801, USA. [5] Department of Biochemistry and Biophysics, Texas A&M University, College Station, Texas 77843, USA. [6] State Key Laboratory of Crop Genetics and Germplasm Enhancement, College of Agriculture, Nanjing Agricultural University, Nanjing 210095, China. [7] Boyce Thompson Institute, Cornell University, Ithaca, New York 14853, USA. [8] USDA-ARS, Southern Plains Agricultural Research Center, College Station, Texas 77845, USA. [9] Texas Agricultural Experiment Station, Lubbock, Texas 79403, USA. [10] Carnegie Science, Department of Plant Biology, 260 Panama Street, Stanford, California 94305, USA. † Present address: Institute of Molecular Physiology, Heinrich Heine University Düsseldorf and Max Planck Institute for Plant Breeding Research, 50829 Köln, Germany. * These authors contributed equally to this work. Correspondence and requests for materials should be addressed to A.J.B. (email: ajb7@cornell.edu) or to L.S. (email: lshan@tamu.edu).

Cotton (*Gossypium* spp.) is an economically important crop and provides a significant source of fibre, feed, foodstuff, oil and biofuel products worldwide. The cotton genus is composed of at least 45 diploid and 5 tetraploid species[1]. The tetraploid (AD genome) species, including *G. hirsutum* that produces 95% of the world's cotton fibre, is likely derived from a hybridization between an A-genome-like ancestral species resembling *G. arboreum* and a D-genome-like ancestral species resembling *G. raimondii*[2]. The availability of the draft genome sequences for *G. raimondii*, *G. arboreum*, *G. hirsutum* and *G. barbadense* not only provides genetic resources to study the complex genome evolution and polyploidization process, but also lays the foundation for functional genomic approaches to dissect cotton gene functions with a goal to improve its agricultural performance in the face of biotic and abiotic stresses[3–8].

Bacterial blight of cotton (BBC), caused by *Xanthomonas citri* subsp. *malvacearum* (*Xcm*), is among the destructive diseases of cotton[9]. Following epidemics in the 1970s, the disease has occurred sporadically in the United States, but in the past several years re-emerged as a significant yield constraint. There are pressing needs to address the underlying mechanisms of susceptibility and resistance to BBC in cotton. *Xcm* injects effector proteins into plant cells via the type III secretion system to promote pathogenicity in plants. In the presence of corresponding resistance (R) proteins, some of these effectors trigger resistance and function as avirulence proteins. Interestingly, all known pathogenicity and avirulence factors (encoded by so called '*pth*' and '*avr*' genes) of *Xcm* are transcription activator-like (TAL) effectors[10]. TAL effectors functionally resemble eukaryotic transcription factors and upregulate host genes by directly binding to their promoters[11,12]. TAL effectors are highly conserved among different *Xanthomonas* spp., with an N-terminal type III translocation signal, a central repeat region (CRR) and C-terminal nuclear localization signals followed by an acidic activation domain. The proteins differ mainly in the CRR, which consists of 1.5–33.5 copies of near-perfect repeats of 33–34 amino acids. These repeats are conserved with the exception of the 12th and 13th residues of each copy, defined as the repeat variable di-residue (RVD)[13,14]. Each RVD targets a specific nucleotide of the promoters of host genes, creating a code, such that the sequence of RVDs defines the effector-binding element (EBE); each EBE starts, however, with a nearly invariant thymine, which is specified in a yet unclear manner by structures immediately N terminus of the CRR[13–16].

Among the identified TAL effector targets include a group of pepper genes upregulated by AvrBs3 from *X. campestris* pv. *vesicatoria* (*Xcv*)[17,18]. These include the *R* gene *Bs3* and *UPA20*, which encodes a basic helix-loop-helix transcription factor regulating plant cell hypertrophy[19]. *Os8N3/Xa13/OsSWEET11*, essential for reproductive development, is a rice susceptibility (S) gene targeted by TAL effector PthXo1 from *X. oryzae* pv. *oryzae* (*Xoo*)[20]. The *xa13* allele, which is unable to be induced by PthXo1, acts as a recessive *r* gene against *Xoo* infection[21,22]. Notably, four *Os8N3/Xa13/OsSWEET11* homologues, rice *Os11N3/OsSWEET14* and *Xa25/OsSWEET13*, pepper *UPA16* and cassava *MeSWEET10a* are also targeted by TAL effectors from *Xoo*, *Xcv* and *X. axonopodis* pv. *manihotis* (*Xam*), respectively[23–25]. It was hypothesized that *Xanthomonas* spp. induces the expression of *SWEET* genes during infection to transport sucrose to the apoplast, thereby providing the bacteria with a carbon source[20,23,25–30]. TAL effectors from *X. citri* sp. *citri* (*Xcc*), the causal agent for citrus bacterial canker, induce the expression of *CsSWEET1* in citrus, but the *CsSWEET1* does not contribute demonstrably to susceptibility; another TAL effector target, however, *CsLOB1*, does function as an *S* gene,

promoting the characteristic pustule formation and bacterial multiplication[31]. Other characterized TAL effector targets include the rice bacterial blight *R* genes *Xa27* (ref. 32), *Xa23* (ref. 33) and *Xa10* (ref. 34), the bacterial leaf streak of rice *S* gene *OsSULTR3;6* (ref. 35) and the pepper bacterial spot *R* gene *Bs4C* (ref. 36).

Previous work identified at least 10 TAL effectors from the cotton pathogen *Xcm*H1005 and showed that a mutant with a deletion of seven *tal* genes was no longer able to cause observable water-soaking symptoms on Acala44 (Ac44), a BBC susceptible line of cotton. Avrb6 from *Xcm*H1005 was among the first examples of an individual TAL effector important for *Xcm* virulence. An *avrb6* mutant of *Xcm*H1005, *Xcm*H1407 (*Xcm*H1005Δ*avrb6*), showed reduced water-soaking, whereas when expressed in a relatively weak virulent strain *Xcm*N1003, Avrb6 enhances water-soaking[37]. Although it does not affect *in planta* bacterial multiplication, Avrb6 plays a major role in release of bacteria from the leaf interior to the leaf surface during infections[37].

To identify the TAL effectors responsible for virulence and their targets, we sequenced the whole genomes of *Xcm* strains *Xcm*H1005 and *Xcm*N1003 and assembled the full repertoire of TAL effectors. Further, we deployed genome-wide gene expression profiling in cotton coupled with TAL effector DNA-binding code-assisted EBE prediction and identified the clade III sucrose transporter gene *GhSWEET10* as a target of Avrb6. We determined that *GhSWEET10* is a major *S* gene for BBC by using dTALEs to induce it independently of any other possible Avrb6 targets and by silencing it with *Agrobacterium*-mediated virus-induced gene silencing (VIGS). Significantly, other members of the clade III *GhSWEET* genes are strongly induced by different *Xcm* field strains responsible for recent re-emergence of the disease in the southern United States. Our data indicate that cotton GhSWEET10 and likely other SWEET sugar transporters in the same clade play a major role in BBC and suggest that newly evolved or horizontally transferred TAL effectors targeting these genes attribute to the re-emergence of BBC in the field. In addition, a BBC-resistant cotton line carrying *b6*, a genetically complex resistance gene that has not yet been molecularly identified, showed markedly reduced Avrb6-mediated induction of *GhSWEET10*, ostensibly due to promoter polymorphisms outside the EBE. This suggests a novel mechanism for resistance that may represent a genetic determinant of *b6*.

## Results

### Genome and TAL effector sequences of *Xcm*H1005 and N1003.
The strains *Xcm*H1005, a rifamycin-resistant derivative of *Xcm*H, and *Xcm*N1003, a spectinomycin- and rifamycin-resistant derivative of *Xcm*N, are two *Xcm* strains that cause water-soaking on Ac44 cotton[38]. To set the stage for identifying the role that TAL effectors in these strains play in BBC, we generated complete genome sequences using single molecule real-time sequencing (Pacific Biosciences; hereafter 'PacBio sequencing'). The H1005 genome consists of a circular chromosome of 5,212,564 bp and a plasmid, pXcmH[39] of 88,283 bp. The N1003 genome consists of a circular chromosome of 5,218,607 bp and a plasmid, herein designated as pXcmN, of 59,644 bp. The general features of each genome are presented in Table 1. The overall architectures of the two chromosomes are highly similar, with a single, large inversion (Fig. 1a). The plasmids are dissimilar, with only a few, relatively small regions of homology (Fig. 1a). The *tal* genes of the two strains are largely distinct (Fig. 1b and Supplementary Table 2). H1005 encodes 12 TAL effectors with six in the chromosome and six in pXcmH. Ten of these are Avr proteins named previously based on reactions of differential cotton lines to transformants of

| Table 1 \| Genome characteristics of *Xcm*H1005 and *Xcm*N1003. | | |
| --- | --- | --- |
| **Characteristics** | ***Xcm*H1005** | ***Xcm*N1003** |
| Length (bp) | 5,212,564 bp | 5,218,607 bp |
| No. of protein-coding genes | 4,227 | 4,218 |
| GC content | 64.62% | 64.54% |
| No. of tRNA genes | 54 | 54 |
| No. of rRNA operons | 2 | 2 |
| Components | Circular chromosome (5.2 Mb); plasmid (88,283 bp) | Circular chromosome (5.2 Mb); plasmid (59,644 bp) |
| No. of TAL effectors (Chromosome::Plasmid) | 6::6 | 5::4 |
| GenBank accession no. | CP013004 | CP013006 |

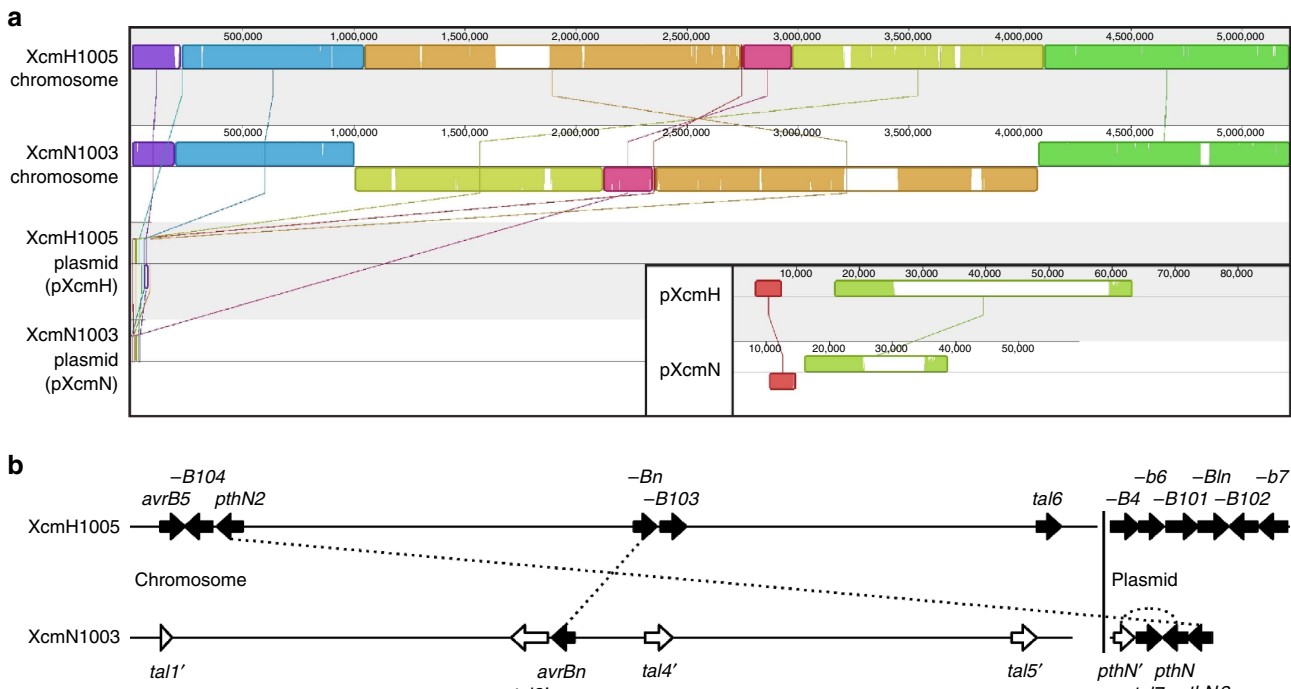

**Figure 1 | Comparison of whole genomes and *tal* genes of *Xcm*H1005 and *Xcm*N1003.** (**a**) Alignment of the *Xcm*H1005 and *Xcm*N1003 genomes, generated using progressiveMAUVE with default parameters[68]. Each genome comprises a single circular chromosome and a single circular plasmid, shown linearized. Coloured, rounded rectangles represent locally collinear blocks (LCB), regions of homology without rearrangement across the aligned sequences, connected by matching coloured diagonal lines. The orientations of the LCB, forward or reverse, are indicated by their position above or below the line, respectively. The height of a column within a block reflects the average similarity relative to the other aligned sequence(s) there (see http://darlinglab.org/mauve/user-guide/viewer.html for details). Inset (bottom right) shows the alignment of the plasmids only, at a larger scale. Horizontal axes show sequence coordinates (bp). MAUVE backbone files giving the exact coordinates of all LCB are provided as Supplementary Data 1 (all molecules aligned) and Supplementary Table 1 (plasmids only). (**b**) The *tal* genes of *Xcm*H1005 and *Xcm*N1003. The genes are represented as block arrows at their relative positions in the chromosome or plasmid (horizontal lines). The arrows are magnified relative to the rest of the genome, but intergenic regions and arrow sizes relative to each other are to scale. Dashed lines connecting two arrows indicate identical encoded RVD sequences. Gene names follow the scheme of ref. 41, except for the indicated *avr* and *pth* genes, named previously[10,38–40]. For label clarity, hypens replace 'avr'. An apostrophe following the gene name (white arrows) indicates that the coding sequence is terminated early due to a frameshift mutation or other coding sequence disruption. Details, including RVD sequences and AnnoTALE designations[53], are given in Supplementary Table 2.

an otherwise-compatible *Xcm* strain carrying corresponding cosmid subclones: AvrB4, AvrB5, Avrb6, Avrb7, AvrB101, AvrB102, AvrB103, AvrB104, AvrBIn and AvrBn[38–40]. Another is PthN2, originally characterized in *Xcm*N1003 (ref. 10). The twelfth is uncharacterized, and we designated it as Tal6$_{XcmH1005}$, following a previously described naming scheme[41]. *Xcm*N1003 harbours nine *tal* genes, five on the chromosome and four on pXcmN. Four on the chromosome and one on the plasmid however are disrupted by a frameshift mutation or a large insertion in the repeat region or the 3′ end of the gene. One of the disrupted genes, *tal1*', on the chromosome, carries only a last repeat and is otherwise complete until an integrase insertion

further downstream. Another of the disrupted genes, *pthN*' on pXcmN, is otherwise identical to *pthN*, an *Xcm*N1003 *tal* gene shown previously to contribute to water-soaking without triggering resistance on commercial US cotton varieties[10]. The intact *pthN* gene is also located on pXcmN, along with *pthN2*, which also contributes to water-soaking, but less strongly[10]. Another intact *tal* gene, *tal7*$_{XcmN1003}$, also resides on the plasmid. The fourth intact gene is *avrBn* located on the chromosome. Apart from *pthN*', *pthN*, *pthN2* and *avrBn*, no other *Xcm*N1003 *tal* genes show obvious similarity in their RVD sequences to any other *tal* genes in either strain. The divergence in *tal* gene content between the two strains, both with respect to RVD composition

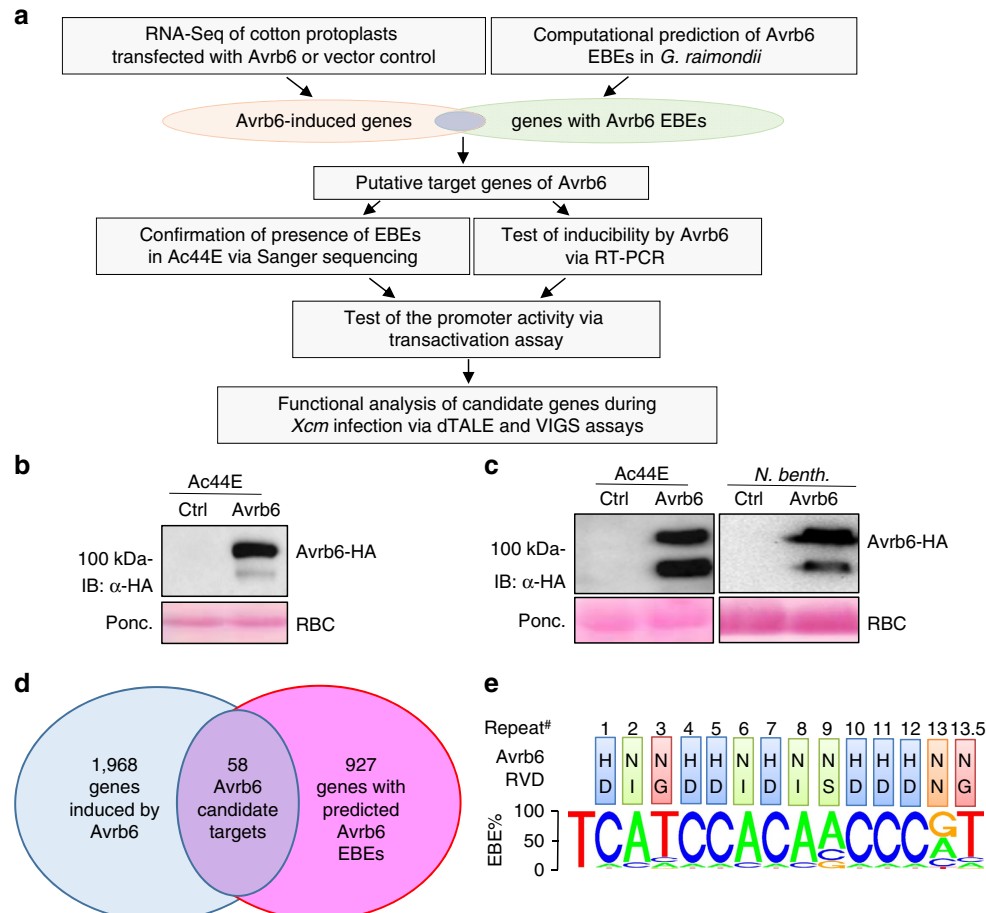

**Figure 2 | Transcriptome profiling coupled with EBE prediction reveals candidate target genes of Avrb6. (a)** Schematic diagram of experimental design to identify Avrb6 target genes in cotton. (**b**) Avrb6 protein expression in cotton protoplasts. Cotton protoplasts isolated from Ac44E were transfected with *avrb6-HA* or an empty vector as a control (Ctrl). Samples were collected 12 h after transfection and subjected to immunoblotting with α-HA antibody (top panel). Ponceau S. staining (Ponc.) of total protein served as the protein loading control; RuBisCo (RBC) is shown (bottom panel). (**c**) Avrb6 protein expression in cotton and *N. benthamiana*. Cotyledons from 2-week-old Ac44E cotton and leaves of 4-week-old *N. benthamiana* were infiltrated with *Agrobacterium* carrying *35S::avrb6-HA* or an empty vector control (Ctrl). Inoculated tissues were collected at 48 hpi and subjected to immunoblotting. B and C were repeated three times with similar results. (**d**) Venn diagram of Avrb6-induced genes and genes with Avrb6 EBEs in *G. raimondii* genome. (**e**) The predicted EBE of Avrb6 RVDs. Coloured boxes on the top panel display the RVD of each repeat of Avrb6. Bottom panel indicates the relative frequencies of RVD associations with the four nucleotides for each repeat.

and genomic location of orthologs, is striking, which is in line with the fact that they were isolated from geographically distant regions[38].

**Candidate targets of Avrb6.** Avrb6 is a TAL effector in *Xcm*H1005 that causes strong water-soaking in the cotton line Ac44 (ref. 37). In this study, we used the Ac44E-tetraploid genotype, which is a single plant selection from Brinkerhoff's original Ac44 parent[42]. Ac44E has similar morphology to Ac44 except it has more abundant flowers and is slightly more susceptible to BBC than Ac44 (ref. 42). To identify the cotton genes that are specifically activated by Avrb6, we used an integrative approach involving a whole-genome RNA-sequencing (RNA-Seq) analysis of protoplasts of Ac44E expressing Avrb6 and computational prediction of Avrb6 EBEs in cotton (Fig. 2a). Compared to the whole plant-pathogen infection assay, the homogenous protoplast mesophyll cell system expressing the individual bacterial effector circumvents the complication of multiple pathogen effectors and elicitors that could spontaneously activate or suppress a large number of host genes. We cloned the full-length coding sequence of Avrb6 from *Xcm*H1005 into

a plant expression vector with an HA-epitope tag at the C terminus under the control of the *35S* CaMV promoter. The sequence was confirmed via Sanger sequencing (Supplementary Fig. 1). An immunoblot analysis with an α-HA antibody detected a major polypeptide slightly above the 100 kDa marker, which matches the predicted molecular mass of 108 kDa for Avrb6 (Fig. 2b). We further subcloned the *35S::avrb6-HA* expression cassette into a binary vector and performed *Agrobacterium*-mediated transient expression assays in Ac44E cotton and *Nicotiana benthamiana*. As shown in Fig. 2c, we observed a similar result as in cotton protoplasts with a major polypeptide being detected at ∼108 kDa. Next, we performed RNA-Seq analysis with cotton protoplasts 12 h after transfection with the *avrb6* construct or a vector control. Approximately 39–50 million raw read pairs and 36–46 million cleaned read pairs were obtained for each sample, which corresponds to ∼450× coverage of 77,267 annotated protein-coding transcripts[1]. Using a twofold expression change and an adjusted *P* value <0.05 as cutoffs, and using the *G. raimondii* genome sequence as a reference (the tetraploid genome sequence was not available at the time), we identified 2,026 genes that were induced by Avrb6 (Fig. 2d and

**Table 2 | The list of top 10 candidate target genes of Avrb6 in cotton based on their inducibility by Avrb6 and the probability of Avrb6 EBEs.**

| Gene ID | Gene name | Induction fold | EBE prob. | Annotation |
|---|---|---|---|---|
| Gorai.008G209000.1 | *GhSWEET10D* | 5973.1 | 1.0 | Bidirectional sugar transporter N3 |
| Gorai.007G067700.1 | *Gh067700* | 29.2 | 0.97 | Unknown protein |
| Gorai.010G056300.1 | *GhKBS1* | 9.6 | 0.96 | Kinase family protein |
| Gorai.009G327300.1 | *GhHLH1* | 4.0 | 0.89 | Basic helix-loop-helix TF |
| Gorai.008G047400.1 | *GhMDR1* | 7.5 | 0.71 | Multi-drug resistance ABC transporter |
| Gorai.010G246000.1 | *Gh2460* | 5.6 | 0.65 | Unknown protein |
| Gorai.007G152300.1 | *GhPPR1* | 5.2 | 0.64 | Pentatricopeptide repeat-containing protein |
| Gorai.008G222500.1 | *GhCYP1* | 471.6 | 0.59 | Cytochrome P450 |
| Gorai.001G131000.2 | *GhZFP1* | 12.3 | 0.51 | Zinc finger family protein |
| Gorai.009G045100.2 | *GhGLY1* | 12.8 | 0.5 | Glycine dehydrogenase |

Supplementary Data 2). To identify genes with potential EBEs for Avrb6, we used the TALE-NT 2.0 Target Finder tool to examine all gene promoter sequences in the *G. raimondii* genome, using the standard score ratio cutoff of 3.0 (Fig. 2e). The list of genes displaying one or more putative EBEs was then intersected with the list of genes upregulated by Avrb6 to produce a list of candidate of Avrb6 targets (Table 2 and Supplementary Data 2). These candidates were ranked by the probability that the predicted EBE is functional by using a machine-learning algorithm based on the output of TALE-NT 2.0 and the genomic context of the predicted EBEs[35]. Interestingly, among the top candidates, Gorai.008G209000.1, a homologue of the SWEET sucrose transporter genes that serve as *S* genes in rice and cassava, was induced ∼6,000-fold by Avrb6 and has an EBE with a probability of 1 (Table 2). Other top candidates include Gorai.007G067700.1, Gorai.010G056300.1, Gorai.009G327300.1 and Gorai.008G047400.1. We then used the gene IDs from *G. raimondii* to identify their homologues in *G. hirsutum*. The gene IDs of the homologues identified in *G. hirsutum* are Gh_D12G1898 for Gorai.008G209000.1, Gh_D11G0631 for Gorai.007G067700.1, Gh_D06G0459 for Gorai.010G056300.1, Gh_D05G2954 for Gorai.009G327300.1 and Gh_D12G0420 for Gorai.008G047400.1. These genes were named *GhSWEET10D* (see below for explanation), *Gh067700*, *GhKBS1*, *GhHLH1* and *GhMDR1*, respectively, based on their predicted gene annotation.

**Direct induction of targets by Avrb6.** To independently determine the induction of the candidate genes, we performed reverse transcription (RT)–PCR assays using Ac44E cotton protoplasts transfected with Avrb6 or with an empty vector as control. The top five candidates, *GhSWEET10*, *Gh067700*, *GhKBS1*, *GhHLH1* and *GhMDR1*, were strongly induced by Avrb6 compared to the vector control (Fig. 3a). We further confirmed induction in cotton leaves by inoculating *Xcm* strains with or without *avrb6*. HM2.2S is derived from H1005 with a deletion of at least seven *tal* genes including *avrb6*, while HM2.2S (*avrb6*) is HM2.2S carrying *avrb6* in plasmid vector pUFR127 (ref. 10). H1005Δ*avrb6* is an *avrb6* mutant derivative of *Xcm*H1005 (ref. 37). As previously reported, *avrb6* in different *Xcm* strains induced strong water-soaking on Ac44E cotton[37] (Fig. 3b). We observed that water-soaking was only induced by the *Xcm* strains carrying *avrb6* (HM2.2S (*avrb6*), HM2.2S (*placZ*::*avrb6*)— a strain that contains the plasmid pUFR135 which expresses *avrb6* driven by the *lacZ* promoter, H1005 (*placZ*::*avrb6*) and H1005). As shown in Fig. 3c, the induction of *GhSWEET10*, *Gh067700*, *GhKBS1* and *GhMDR1* by the strains carrying *avrb6*, HM2.2S (*avrb6*) and H1005, was stronger than that by strains lacking *avrb6*, HM2.2S and H1005Δ*avrb6*, at 12- and

24-h post inoculation (hpi). We could not amplify the *GhHLH1* gene in cotton likely due to low transcript level. To explore whether *GhSWEET10*, *Gh067700*, *GhKBS1* and *GhMDR1* are directly induced by Avrb6, we added the eukaryotic protein synthesis inhibitor cycloheximide (CHX; 50 μM final concentration) to the *Xcm* suspensions and inoculated into Ac44E cotton leaves. Each gene was still induced by strains carrying *avrb6*, H1005 and HM2.2S (*avrb6*), in the presence of CHX at 24 hpi (Fig. 3d), suggesting that new protein biosynthesis is not required for induction of these genes and that they are direct targets of Avrb6. The effectiveness of CHX in suppressing protein synthesis in this context was confirmed by its suppression of Avrb6 protein expression in a separate *Agrobacterium*-mediated transient expression assay in Ac44E cotton (Supplementary Fig. 2). To examine potential polymorphisms in these gene promoters between the sequenced cotton and Ac44E cotton used in this study, we amplified each promoter from Ac44E for sequencing. This revealed that each of the promoters possesses an Avrb6 EBE except *Gh067700*, which has a 1 bp deletion in the EBE in Ac44E (Supplementary Fig. 3). To confirm whether Avrb6 activates the genes by targeting the EBEs, we amplified ∼800 bp upstream of the translational start site of each gene, cloned each of these fragments upstream of a luciferase reporter gene (*LUC*) and performed a cotton protoplast-based transactivation assay (Fig. 3e). Compared to an empty vector control, co-transfection of Avrb6 strongly activated *pGhSWEET10D::LUC* with about 110-fold induction, and also activated *pGhKBS1::LUC* and *pGhMDR1::LUC* with about 15–20-fold induction (Fig. 3f). Significantly, these promoters were not activated by PthN, another *Xcm* TAL effector with an RVD sequence distinct from Avrb6, indicating specificity of activation by Avrb6 (Fig. 3f). Consistent with the observation that *Gh067700* in cotton Ac44E carries a nucleotide deletion in Avrb6 EBE, *pGh067700::LUC* was not activated by Avrb6. In addition, *pGhHLH1::LUC* was not significantly activated by Avrb6, though its promoter possesses a predicted Avrb6 EBE. Since the promoter of *GhSWEET10D* was strongly induced by Avrb6, we tested the role of the EBE by mutating two nucleotides to create a version of *pGhSWEET10D::LUC* with a 'CA' to 'GG' substitution in the EBE (*mEBE*; Fig. 3g). Avrb6 strongly activated *pGhSWEET10D::LUC* carrying the wild-type EBE, but not the *mEBE* (Fig. 3h). Taken together, these data indicate that Avrb6 directly and specifically activates *GhSWEET10D*, and likely *GhKBS1* and *GhMDR1*, and that the EBE is required for Avrb6-induced expression of *GhSWEET10D*.

**GhSWEET10 activation using a dTALE induces water-soaking.** We generated dTALEs to specifically activate *GhSWEET10D*,

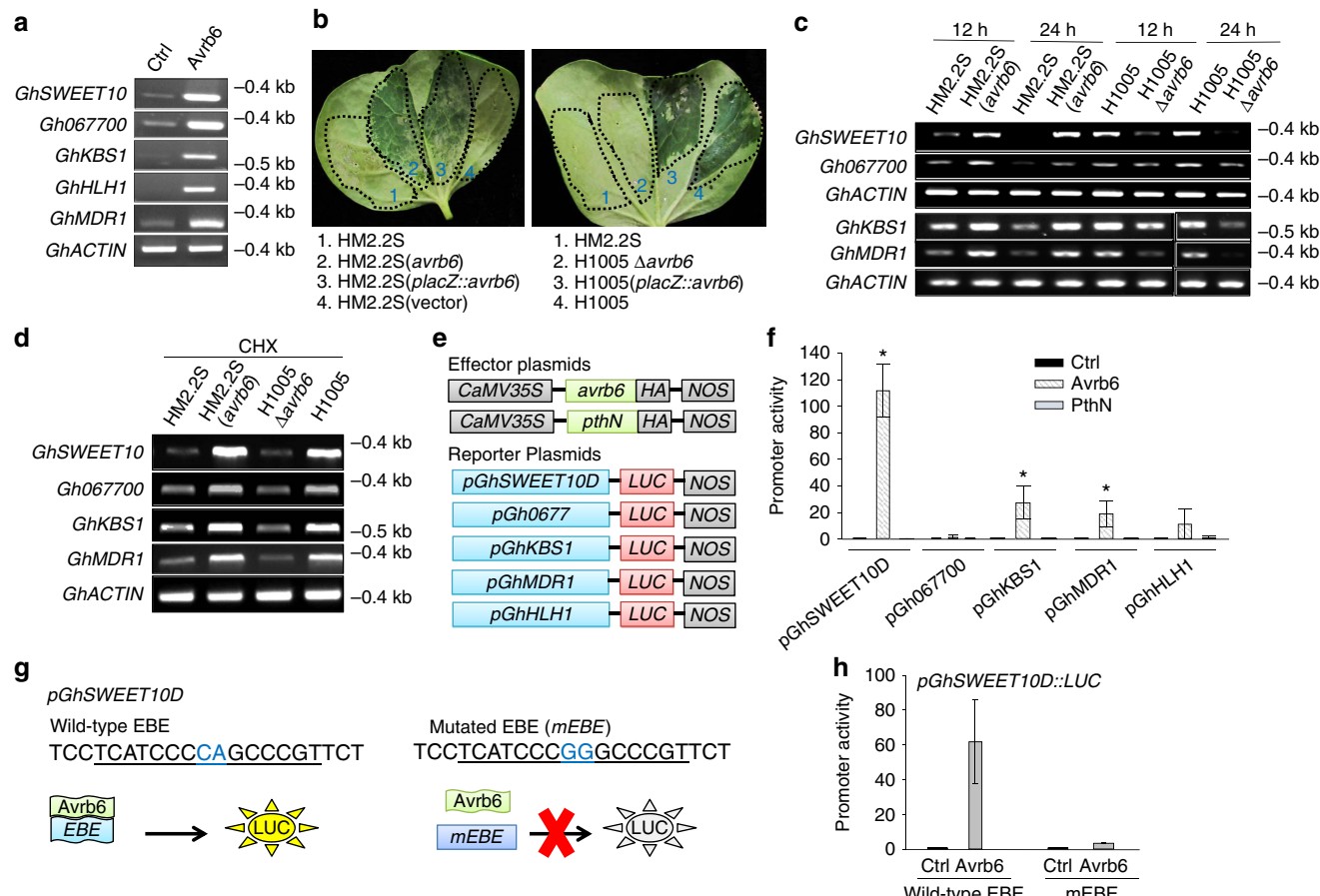

**Figure 3 | Avrb6 directly induces cotton gene transcription in the absence of protein synthesis.** (**a**) RT–PCR analysis of Avrb6 upregulated genes in cotton protoplasts. Cotton protoplasts of Ac44E were transfected with *avrb6-HA* or a vector control (Ctrl) and incubated for 12 h before RNA isolation. *GhACTIN* was used as an internal control. (**b**) Avrb6 contributes to water-soaking development in cotton. Cotyledons from 2-week-old Ac44E cotton were syringe-inoculated with different *Xcm* strains at OD$_{600}$ = 0.1 and photographed 4 days after infiltration. Inoculation areas are indicated by dotted lines. (**c**) RT–PCR analysis of Avrb6 upregulated genes in cotton upon *Xcm* infection. Cotyledons from 2-week-old Ac44E cotton were syringe-inoculated with different *Xcm* strains. Tissues were collected at 12 and 24 hpi. (**d**) RT–PCR analysis of Avrb6 upregulated genes in the presence of CHX. Cotyledons from 2-week-old cotton were syringe-inoculated with different *Xcm* strains in 50 μM CHX. Tissues were collected at 24 hpi for RT-PCR. (**e**) Schematic diagram of the effector and reporter constructs. The reporter construct contains an expression cassette with a *LUC* reporter gene under the control of a candidate gene promoter. The effector construct contains either Avrb6 or PthN with an HA-epitope tag under the control of the CaMV *35S* promoter. (**f**) Transcriptional activity of Avrb6 in cotton protoplasts. Protoplasts were co-transfected with a reporter construct and *avrb6*, *pthN* or a vector control (Ctrl), and were collected 12 h after transfection. *UBQ10-GUS* was included in the transfections as an internal control. The luciferase activity was normalized with GUS activity. The data are shown as mean ± s.d. (*n* = 3) from three independent repeats. Asterisks indicate significant difference using two-tailed *t*-test (*P* < 0.05). (**g**) Schematic diagram of the transactivation assay of wild-type and mutated EBE (mEBE) in *pGhSWEET10D* in response to Avrb6. The nucleotide sequence containing the putative EBE is shown and the two nucleotides that were mutated are highlighted in blue. (**h**) Transcriptional activity of *GhSWEET10D* with wild type and mEBE in response to Avrb6. Cotton protoplasts were co-transfected with *pGhSWEET10D::LUC* carrying wild type or mEBE and Avrb6 or a vector control (Ctrl). The data are shown as mean ± s.d. (*n* = 3) from three independent repeats. The above experiments were repeated three times with similar results.

*GhKBS1* or *GhMDR1* via EBEs distinct from the corresponding Avrb6 EBEs, introduced these into the *Xcm* HM2.2S strain, which lacks *avrb6*, and performed disease assays in cotton. The dTALEs were expressed from the low-copy, broad-host range vector, pKEB1, a derivative we constructed from pUFR047 (ref. 38) that carries a gateway cloning cassette (Fig. 4a). Significantly, HM2.2S transformants carrying the dTALE targeting *GhSWEET10D* (dTALE1), but not those targeting *GhMDR1* (dTALE3) or *GhKBS1* (dTALE4), or the vector control, induced water-soaking in Ac44E cotton, similarly to *Xcm*H1005 (Fig. 4b). To determine if the dTALEs used indeed activate the transcripts of their respective target genes, we performed RT–PCR analysis. As shown in Fig. 4c, *GhSWEET10D* and *GhMDR1* were induced by the cognate dTALEs. These data indicate that activation of

*GhSWEET10D* on *Xcm* infection contributes to water-soaking development, thus identifying *GhSWEET10D* as a relevant target of Avrb6 in cotton and an *S* gene. We did not detect the induction of *GhKBS1* by its dTALE, and cannot conclude whether GhKBS1 plays a role in BBC.

**Avrb6 targets *GhSWEET10* in both A and D genomes of cotton.** The tetraploid *G. hirsutum* has two subgenomes, A and D. The above studies of *GhSWEET10D* were mainly based on the D genome of *G. raimondii* and the corresponding D subgenome of *G. hirsutum*. We examined if Avrb6 also targets *SWEET10* in the A-genome of *G. arboreum*, which is closest to the progenitor species of the A subgenome of *G. hirsutum*. We identified a

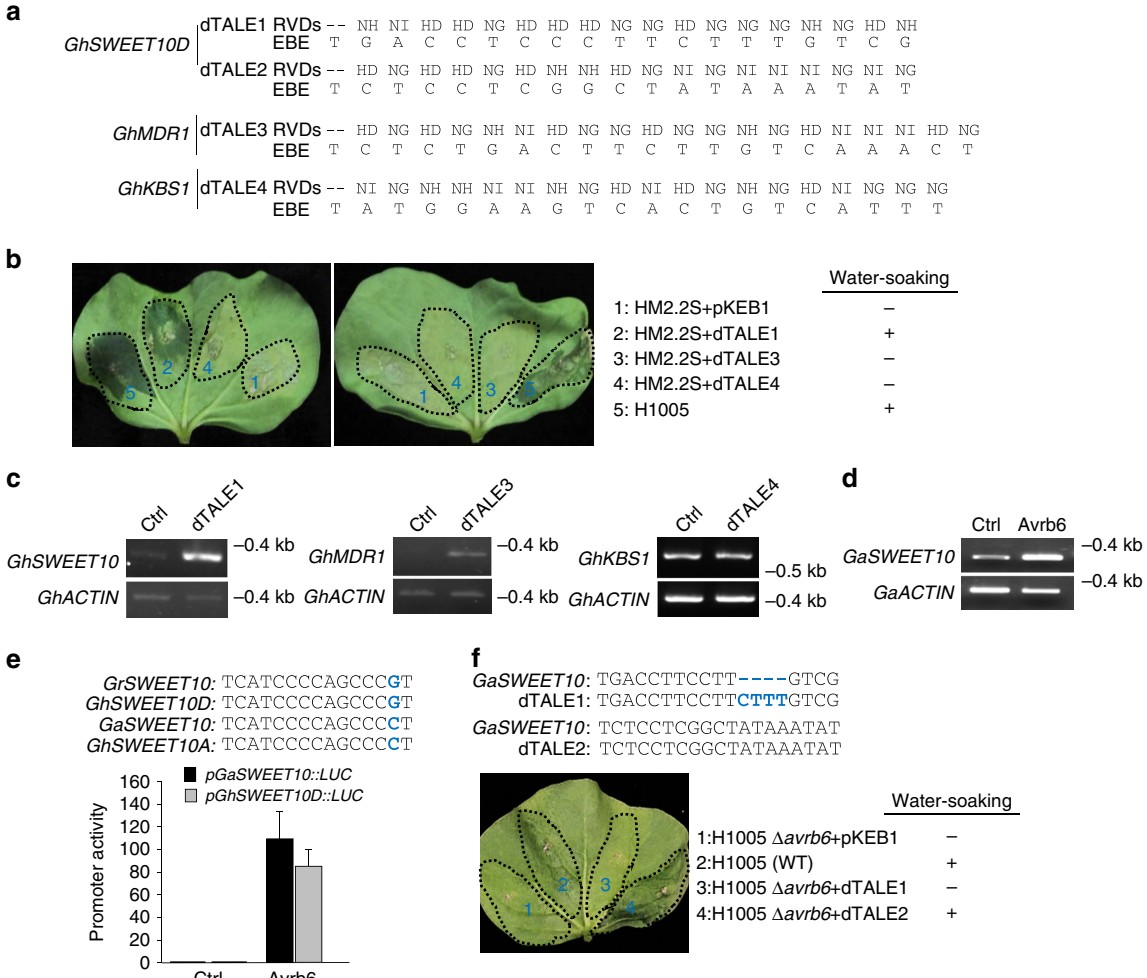

**Figure 4 | The dTALE-activating GhSWEET10 induces water-soaking in cotton.** (**a**) RVDs of dTALEs and the corresponding targeted EBE sequences in the *G. hirsutum* genome. (**b**) *Xcm* HM2.2S expressing a dTALE uniquely targeting *GhSWEET10D* induces water-soaking in cotton. Cotyledons from 2-week-old Ac44E plants were syringe-inoculated with different *Xcm* strains at $OD_{600} = 0.1$ and photographed at 4 dpi. The table displays the presence ( + ) or absence ( − ) of water-soaking. (**c**) RT–PCR analysis of *GhSWEET10*, *GhKBS1* and *GhMDR1* targeted by their respective dTALEs. Two-week-old cotyledons of Ac44E were syringe-inoculated with *Xcm* strains HM2.2S ($OD_{600} = 0.1$) containing different dTALEs. Inoculated tissues were collected at 24 hpi for RT–PCR analysis. (**d**) *GaSWEET10* is induced by Avrb6 in *G. arboreum* protoplasts. Protoplasts isolated from *G. arboreum* were transfected with Avrb6 or an empty vector control (Ctrl) and samples were collected 12 h after transfection for RT–PCR analysis. (**e**) Transactivation assay of *GaSWEET10* and *GhSWEET10D* promoters in response to Avrb6 in cotton protoplasts. Alignment of Avrb6 EBEs reveals a polymorphism between *GhSWEET10D* and *GaSWEET10* promoters, which were highlighted in blue. Protoplasts isolated from commercial cotton variety FM706V were co-transfected with *pGaSWEET10::LUC* or *pGhSWEET10D::LUC* and Avrb6 or a vector control (Ctrl). The data are shown as mean ± s.d. ($n = 3$) from three independent repeats. (**f**) The dTALE matching the *GaSWEET10* promoter causes water-soaking on *G. arboreum*. The EBE sequences of two dTALEs targeted to different regions of the *GaSWEET10* promoter are shown. dTALE2 with an EBE sequence perfectly matching the *GaSWEET10* promoter but not dTALE1 with a partially matching EBE sequence restores *Xcm*-mediated water-soaking on cotton. Cotyledons from 2-week-old *G. arboreum* were syringe-inoculated with different *Xcm* strains at $OD_{600} = 0.1$ and photographed at 4 dpi. The above experiments were repeated three times with similar results.

homologue of GhSWEET10D, named GaSWEET10, which bears 98% similarity to GhSWEET10D at the amino-acid level. We expressed Avrb6 in *G. arboreum* cotton protoplasts and determined whether *GaSWEET10* was induced by Avrb6. RT–PCR analysis showed that *GaSWEET10* was induced by Avrb6 (Fig. 4d). To determine whether Avrb6 could specifically activate the promoter of *GaSWEET10*, we cloned the promoter, *pGaSWEET10*, upstream of the luciferase gene for a transactivation assay. As shown in Fig. 4e, Avrb6 activated *pGaSWEET10::LUC* to a similar level as Avrb6 activation of *pGhSWEET10D::LUC*. Notably, an alignment of the Avrb6 EBE from *pGhSWEET10D* with the one from *pGaSWEET10* revealed them to be identical except for a single nucleotide polymorphism near the 3′ end of the EBE (position 14, counting the initial

thymine), where mismatches are known to be generally tolerated[43]. Additionally, an alignment of protein-coding sequences revealed that GrSWEET10 from *G. raimondii* is identical to D subgenome GhSWEET10D, whereas GaSWEET10 from *G. arboreum* is identical to A subgenome GhSWEET10A (Supplementary Fig. 4). Similarly, the promoter sequence, including the EBE, of *pGhSWEET10A* from tetraploid Ac44E was more closely related to *pGaSWEET10* from *G. arboreum* versus *pGrSWEET10* from *G. raimondii* (Supplementary Fig. 4b). We used the dTALE approach to determine if *GaSWEET10* is also a direct target of Avrb6 that causes water-soaking. As shown in Fig. 4f, the dTALE2 construct, but not the dTALE1 construct specific to *pGhSWEET10D* or an empty vector introduced in H1005Δ*avrb6*-induced water-soaking in *G. arboreum*. Notably,

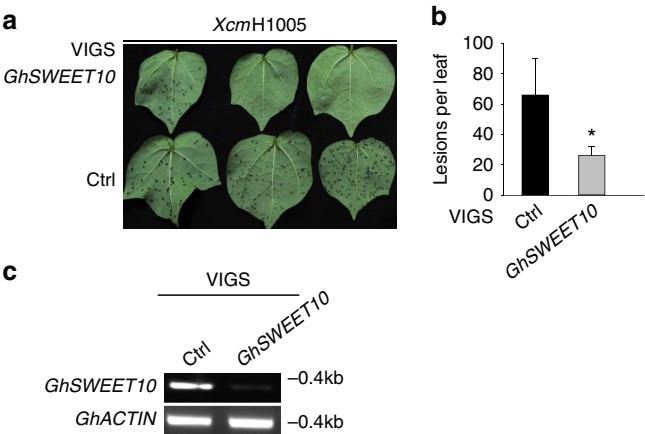

**Figure 5 | Silencing of *GhSWEET10* in cotton reduces water-soaking caused by *Xcm* infection.** (**a**) *Xcm*-mediated water-soaking lesion development on cotton plants upon VIGS. Cotyledons from 2-week-old Ac44E plants were syringe-infiltrated with *Agrobacterium* carrying a VIGS construct to silence *GhSWEET10* (VIGS-*GhSWEET10*) or a GFP vector control (VIGS-Ctrl). Three weeks later, plants were vacuum infiltrated with *Xcm*H1005 at $OD_{600} = 0.01$. Images were taken 2 weeks after inoculation. (**b**) Quantitative analysis of *Xcm*-mediated water-soaking lesions on cotton plants upon VIGS. Lesions were counted on the second true leaf of each VIGS plant (at least 10 plants were inoculated for each construct in each repeat). The data are shown as mean ± s.e. ($n = 10$). The asterisk indicates a significant difference with a Student's *t*-test ($P < 0.05$) when compared with the control. (**c**) RT–PCR analysis of *GhSWEET10* expression in cotton plants upon VIGS. VIGS assays were done similarly as in **a** and 3 weeks later, the second true leaf was collected for RT–PCR analysis before *Xcm* inoculation. *GhACTIN* was used as an internal control. The above experiments were repeated three times with similar results.

H1005 induced relatively weak water-soaking on *G. arboreum* compared to *G. hirsutum*, likely due to *G. arboreum* carrying some strain-specific resistance as noted previously[44]. Taken together, these results indicate that Avrb6 can directly target SWEET10 in both the A and D genomes to contribute to the development of BBC.

**GhSWEET10 silencing reduces water-soaking caused by *Xcm*.** Having determined that *GhSWEET10* mediates Avrb6-induced water-soaking, we subsequently investigated whether *GhSWEET10* is genetically required for water-soaking during *Xcm* infection in cotton. We employed the *Agrobacterium*-mediated VIGS system to silence *GhSWEET10* and examined BBC development after subsequent inoculation of *Xcm*. The VIGS construct was designed to silence both *GhSWEET10A* and *GhSWEET10D* because of sequence conservation. There was no apparent difference in plant growth between the VIGS-*GhSWEET10* plants and plants inoculated with the GFP control (VIGS-Ctrl) up to 3 weeks after VIGS (Supplementary Fig. 5). Three weeks after *Agrobacterium* infiltration, the silenced plants were vacuum infiltrated with *Xcm*H1005. Compared to the control, the VIGS-*GhSWEET10* plants consistently showed fewer water-soaked lesions after *Xcm*H1005 infection (Fig. 5a,b). RT–PCR analysis confirmed reduced induction of *GhSWEET10* in VIGS-*GhSWEET10* plants compared to VIGS-Ctrl plants (Fig. 5c). Taken together, the data indicate that silencing of *GhSWEET10* in cotton reduces the development of water-soaking following *Xcm* inoculation, providing genetic evidence that *GhSWEET10* is functionally required for normal BBC symptom development upon infection by *Xcm* strains carrying *avrb6*.

**GhSWEET10D encodes a functional sucrose transporter.** Since *GhSWEET10* belongs to the *SWEET* gene family, we examined whether GhSWEET10 functions as a sugar transporter. To measure activity, we transiently expressed the coding sequence of GhSWEET10D along with a Förster resonance energy transfer (FRET) sucrose sensor in human embryonic kidney T293 (HEK293T) cells[45]. Compared to the vector control, expression of GhSWEET10D enabled HEK293T cells to accumulate sucrose, which was detected as a negative ratio change, to a level comparable to the well-characterized *Arabidopsis* transporter AtSWEET11 (Fig. 6a). In addition, when we transiently expressed *GhSWEET10D* under the control of the *35S* CaMV promoter in *N. benthamiana*, the sucrose concentration of the apoplastic fluids from the *35S::GhSWEET10D*-inoculated leaves was higher than that from the vector control-inoculated leaves (Supplementary Fig. 6a), indicating that GhSWEET10D is able to transport sucrose *in planta*. Interestingly, *GhSWEET10D* did not restore the growth defect of yeast EBY4000, a hexose transport-deficient yeast strain, on the medium supplemented with 2% glucose or 2% fructose as the sole carbon source, indicating that GhSWEET10D does not mediate efficient fructose or glucose transport (Supplementary Fig. 6b). Taken together, the data suggested that *GhSWEET10D* encodes a functional sucrose transporter.

**GhSWEET10D is a member of a large gene family in cotton.** In bacterial blight of rice, several closely related *SWEET* genes can each serve as an *S* gene[28]. To better understand the relationship within the SWEET family in cotton, especially which SWEETs are most closely related to GhSWEET10, the full-length protein sequence of GhSWEET10D was used as the query for a BLASTp analysis against the *G. hirsutum* NBI protein database (www.cottongen.org/tools/blast). The search identified 50 additional GhSWEETs, for a total of 51 SWEETs in *G. hirsutum*, with each possessing the conserved domains found in the AtSWEET superfamily in *Arabidopsis* (Supplementary Figs 7 and 8). Based on the well-characterized phylogenetic analysis of AtSWEETs, we generated a phylogenetic tree and classified the GhSWEET family into four clades with nine in clade I, 14 in clade II, 18 in clade III and 10 in clade IV. The individual members of GhSWEETs are named using the tree described by Eom *et al.*[46], as a reference where GhSWEET1-3 are in clade I, GhSWEET4-8 are in clade II, GhSWEET9-15 are in clade III and GhSWEET16-18 are in clade IV. Considering the gene duplication and polyploidy in tetraploid cotton, lowercase and uppercase letters are used, respectively, to distinguish closely related members. If multiple members are co-orthologous to the closest AtSWEET, lowercase letters (a–c) are added after the numbers to differentiate genes in the same subclade. An uppercase 'A' is further added to indicate the gene is from the A subgenome, or an uppercase 'D' for the D subgenome. For example, there appear to be two pairs of GhSWEETs that are closely related to AtSWEET1 as shown in the phylogenetic tree. Therefore, we distinguish these by naming one pair as GhSWEET1a and another pair as GhSWEET1b. Each of these pairs is further named according to their subgenome locations as GhSWEET1aA and GhSWEET1aD, and GhSWEET1bA and GhSWEET1bD, respectively (Fig. 6b and Supplementary Fig. 7). Like the *SWEET* genes in rice and cassava that serve as *S* genes, *GhSWEET10* is a member of the clade III family (Fig. 6c).

**GhSWEET10 induction by Avrb6 is reduced in *b6* cotton.** Avrb6 causes severe water-soaking in the susceptible Ac44E cotton line. However, in the near-isogenic resistant line Acb6,

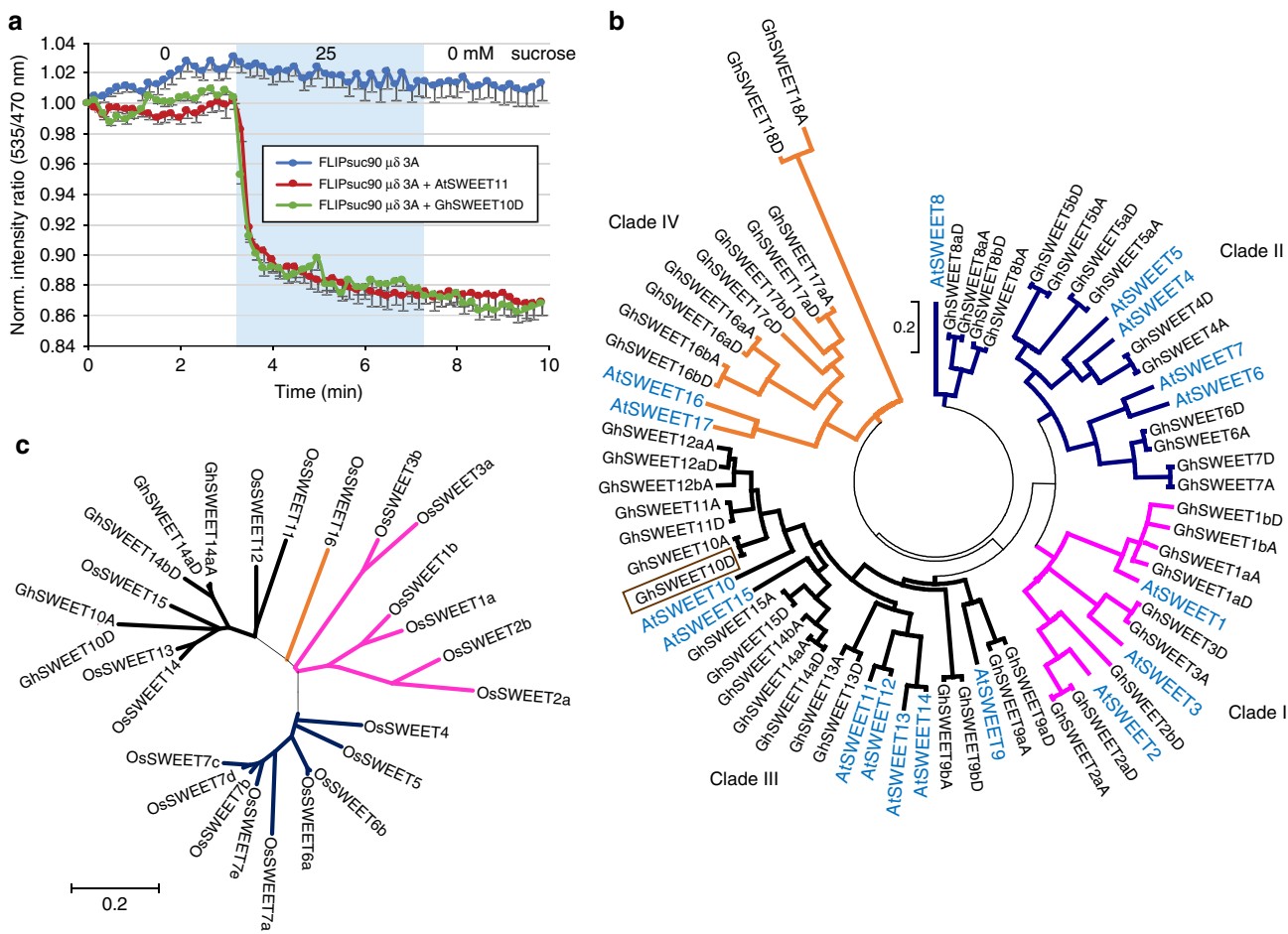

**Figure 6 | *GhSWEET10D* encodes a functional clade III sucrose transporter. (a)** Detection of GhSWEET10D transporter activity in HEK293T cells using the sucrose sensor FLIPsuc90μδ3A. Sucrose transporter activity was assayed by co-expressing GhSWEET10D with the cytosolic FRET sucrose sensor FLIPsuc90μδ3A in HEK293T cells. Blue circles correspond to cells expressing the sensor alone; red circles and green circles correspond to cells co-expressing the sensor with AtSWEET11 or GhSWEET10D, respectively. The cyan block indicates duration of perfusion with 25 mM sucrose. Accumulation of sucrose is reported as a negative ratio change (mean − s.e.m.; n > 10). Experiments were repeated four times with similar results. **(b)** Phylogenetic analysis of GhSWEET proteins from *G. hirsutum*. AtSWEET proteins from *Arabidopsis thaliana* were included to represent different clades of SWEET superfamily. The phylogenetic tree was made using the neighbour-joining method in MEGA version 6.06. The evolutionary distances were computed using the Poisson correction method and are in the units of the number of amino acid substitutions per site[69]. *GhSWEET10D*, which was used as the query, is highlighted with a brown box. **(c)** Phylogenetic analysis of GhSWEET10, GhSWEET14a, GhSWEET14b and the OsSWEET proteins from *Oryza sativa*. The phylogenetic tree was made as described in **b**. The clades of the SWEET family were colour coded as follows: clade I = pink, clade II = blue, clade III = black and clade IV = orange.

which harbours the apparently genetically complex BBC recessive resistance locus *b6*, Avrb6 causes the resistance-associated hypersensitive response (HR) instead (ref. 38 and Supplementary Fig. 9a,b). We examined whether *b6*-mediated resistance is in part due to the inability of Avrb6 to activate *GhSWEET10* in Acb6. When we expressed Avrb6 in Ac44E and Acb6 protoplasts, respectively, the induction of *GhSWEET10* by Avrb6 was significantly lower in Acb6 than that in Ac44E (Fig. 7a). We further compared *GhSWEET10* induction in Ac44E and Acb6 cotton plants inoculated with different *Xcm* strains with or without *avrb6*. Both Ac44E and Acb6 plants showed increased induction of *GhSWEET10* on inoculation with HM2.2S carrying *avrb6* compared to the plants inoculated with HM2.2S (Fig. 7b). However, the induction was significantly reduced in Acb6 plants relative to Ac44E plants. A similar trend was observed when H1005 was used to inoculate both lines (Fig. 7b). To examine whether the reduced induction by Avrb6 in Acb6 could be due to polymorphisms at the *GhSWEET10D* EBE, we sequenced the *GhSWEET10D* promoter region from Acb6. The EBE is perfectly conserved, but in Acb6, there is a SNP immediately adjacent to the 5′ of the EBE and a 2 bp insertion surrounded by six additional SNPs ∼20 bp downstream of the EBE (Supplementary Fig. 9c). We cloned the promoter of *GhSWEET10D* from Acb6 in front of the luciferase reporter for a transactivation assay in cotton protoplasts. Similar to what we observed for the endogenous *GhSWEET10D* gene, the induction of the *pGhSWEET10D_{Acb6}::LUC* reporter by Avrb6 was nearly three folds lower than that of the reporter carrying the promoter from Ac44E (*pGhSWEET10D_{Ac44E}::LUC*) (Fig. 7c). The promoter of *GhSWEET10A* is almost identical in Ac44E and Acb6 (Supplementary Fig. 4b). Thus, the polymorphisms outside the *GhSWEET10D* EBE may impair full induction by Avrb6. In addition, when we inoculated H1005Δ*avrb6 carrying* dTALE2, which activates *pGhSWEET10D* (Fig. 4a,f), into cotyledons of Acb6, the bacterium triggered water-soaking, but not HR in Acb6 (Supplementary Fig. 9d). The data suggest that *GhSWEET10D* may not mediate Avrb6-induced HR and Avrb6 targets another gene for HR in Acb6.

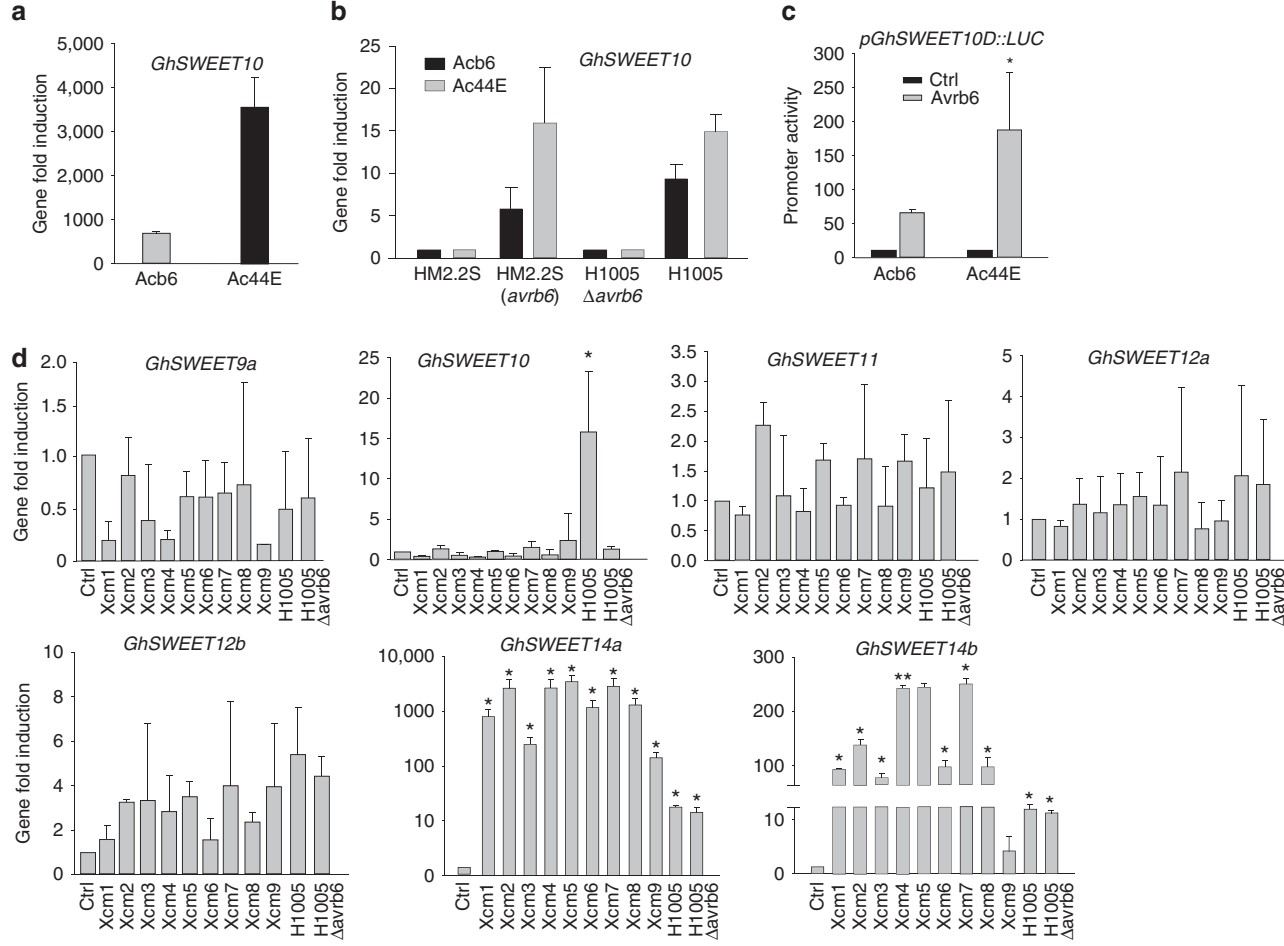

**Figure 7 | Induction of *GhSWEET* genes in different cotton–*Xcm* interactions.** (**a**) *GhSWEET10* induction by Avrb6 in protoplasts of the resistant line Acb6 and the susceptible line Ac44E. Graph displays Avrb6-induced *GhSWEET10* reads per kilobase of transcript per million mapped reads and fold change from RNA-Seq analysis with Ac44E (compatible) and Acb6 (incompatible) cotton lines. (**b**) qRT–PCR analysis of *GhSWEET10* in Ac44E and Acb6 plants upon *Xcm* infections. Two-week-old cotyledons were syringe-inoculated with different *Xcm* strains at $OD_{600} = 0.1$ and tissues were collected at 24 hpi. *GhUBQ1* was used as an internal control. The data are shown as mean ± s.d. ($n = 3$) from three independent repeats. (**c**) Transactivation assay of *pGhSWEET10D::LUC* from Acb6 and Ac44E in response to Avrb6 in cotton protoplasts. The promoter of *GhSWEET10D* was amplified from Acb6 and Ac44E, respectively, and fused with luciferase reporter. Protoplasts of cotton variety FM706V were co-transfected with the reporter construct and Avrb6 or an empty vector control (Ctrl). The data are shown as mean ± s.d. ($n = 3$) from three independent repeats. An asterisk indicates the significant difference using two-tailed *t*-test ($P < 0.05$) between two reporter constructs after Avrb6 induction. (**d**) Induction of clade III *GhSWEET* genes by different *Xcm* isolates. Two-week-old cotyledons were syringe-inoculated with different *Xcm* isolates (*Xcm1-Xcm9*) collected from different locations in Texas, USA at $OD_{600} = 0.1$. Tissues were collected at 24 hpi. *GhUBQ1* was used as an internal control for qRT–PCR analysis. The data are shown as mean ± s.d. ($n = 3$) from three independent repeats. An asterisk indicates significant difference using two-tailed *t*-test ($P < 0.01$) compared to the water control. The above experiments were repeated three times with comparable results.

**Clade III *GhSWEET* responsiveness to different *Xcm* isolates.** Clade III *SWEET* genes have been implicated in disease susceptibility in rice and cassava. We here identified a clade III *SWEET* gene in cotton, *GhSWEET10*, that encodes a sucrose transporter and that also plays a role in disease susceptibility. To examine how general the involvement of clade III *GhSWEET* genes in BBC is, we investigated the transcriptional changes of these genes in response to nine *Xcm* isolates collected from infected cotton fields from different locations in Texas. Notably, these isolates are suspected to be the causal agents of the recent rampant re-emergence of BBC in the United States. However, it has not been determined yet whether these isolates represent a new race. We found that the field isolates were unable to induce *GhSWEET10*. However, all of the field isolates highly induced *GhSWEET14a* (200–4,000-fold) (Fig. 7d). In addition, all of the isolates, except for *Xcm9*, also significantly induced *GhSWEET14b* (20–300-fold). As expected, H1005, but not H1005Δ*avrb6*,

induced *GhSWEET10* by 15-fold. Notably, *GhSWEET14a* and *GhSWEET14b* are clade III *SWEET* genes (Fig. 6c). Taken together, these data suggest that in addition to *GhSWEET10*, *GhSWEET14a* and *GhSWEET14b* are potential *S* genes, and that activating clade III *GhSWEET* genes might be broadly important to the ability of different *Xcm* strains to cause BBC.

## Discussion

BBC caused by *Xcm* is among devastating diseases that limit cotton yields. Despite several TAL effectors being cloned from *Xcm*, no *R* or *S* genes have yet been molecularly identified, mainly because of the large and polyploidy nature of the cotton genome and limited molecular and genomic tools. The approach taken in this study successfully overcame these challenges to establish a role for SWEET-mediated susceptibility in BBC. *GhSWEET10* was strongly induced by Avrb6 during *Xcm*

infection and was identified as an *S* gene for BBC. Characterization of BBC-resistant line Acb6 revealed sequence differences in the *GhSWEET10D* promoter, but not in the EBE, that impair activation by Avrb6, possibly relating to resistance mediated by the *b6* gene. Finally, surveys of different *Xcm* field isolates showed that two additional clade III *GhSWEETs* were highly induced during infection. It appears that the evolution of novel TAL effectors that induce these additional clade III *GhSWEET* members may have contributed to the recent re-emergence of BBC in the United States.

The *SWEET* family members are key *S* genes common to several diseases caused by different *Xanthomonas* species[47]. So far, only clade III *SWEET* genes are targeted by pathogens that use TAL effectors to facilitate disease[46,48]. Members of clade III *SWEET* genes are known to transport sucrose, which likely serves as a carbon source for the pathogen. Interestingly, it was recently shown that two clade III *SWEET* genes in *Arabidopsis*, *AtSWEET13* and *AtSWEET14*, can also transport plant hormone gibberellin to modulate growth and development[49]. In rice, *Xoo* can target three clade III *SWEET* genes to cause susceptibility. *Xoo* TAL effector PthXo1 targets *OsSWEET11* (ref. 20); PthXo2 targets *OsSWEET13* (ref. 27); and AvrXa7 targets *OsSWEET14* (ref. 23). In cassava, *Xam* can use TAL20$_{Xam668}$ to target *MeSWEET10a*, another clade III *SWEET* gene[25]. In at least one case, a host *SWEET* gene is targeted by multiple TAL effectors from different strains of the pathogen: in addition to AvrXa7, *OsSWEET14* is targeted by PthXo3, TalC and Tal5, each from a different *Xoo* strain[23,28,50]. The central role of SWEETs in susceptibility is further supported by our data, as we have identified *GhSWEET10* as a clade III *SWEET* gene in cotton targeted by Avrb6. In addition, we observed that several different *Xcm* field isolates strongly induce two additional clade III *GhSWEETs*, *GhSWEET14a* and *GhSWEET14b*, but not *GhSWEET10*, during infections. Since *Xcm*H1005 is a strain derived from an older race, and since these field isolates were collected within the past year, this strongly suggests that new TAL effectors have evolved in these field isolates to target new *SWEETs* and break resistance. The striking differences of the *tal* gene content and the localization of the *pthN2* gene on chromosome or plasmid in *Xcm*H1005 and *Xcm*N1003 are consistent with the notion that TAL effectors in *Xcm* may evolve rapidly. However, it is also possible that TAL effectors from these recent strains can activate another target, either another *SWEET* gene or a gene in different family, to cause susceptibility. It was revealed that TalC from *Xoo* can still cause disease on rice, despite having its corresponding EBE from *OsSWEET14* artificially mutated, suggesting that preventing TalC-mediated *SWEET14* activation alone is not sufficient to prevent susceptibility[51]. Additionally, even though *GhSWEET10*, *GhSWEET14a* and *GhSWEET14b* reside in clade III, it is still unclear if they function similarly. Additional biochemical analyses are needed to further understand the function of these genes in BBC. Characterizing the spectrum of SWEET-targeting TAL effectors and the relative contributions of their targets to susceptibility, both in the older strains and the recent isolates, is an important future objective.

The CRR of TAL effectors determines their DNA-binding specificity for targeted gene activation in disease or resistance. Inventory of the repeat sequences is essential to predict important TAL effector targets. However, the repeats are generally impossible to assemble from Illumina-sequencing reads and challenge remains even with Sanger sequencing. PacBio sequencing has proven to be an effective approach however[52–54]. We used PacBio sequencing to reveal the whole repertoire of TAL effectors in *Xcm*H1005 and *Xcm*N1003. We identified 12 TAL effectors encoded by H1005 (six in the chromosome and six in the plasmid) and 9 encoded by N1003 (five in the chromosome

and four in the plasmid). In addition to *GhSWEET* genes, another shared class of targets across TAL effectors of different *Xanthomonas* species is transcription factor genes[19,31,55]. In at least one case, the CsLOB1 transcription factor of citrus, and other members of CsLOB1 family also function as *S* genes[31,56]. It will be interesting to determine whether targets of other TAL effectors in H1005 and N1003 similarly include transcription factor genes, or yet novel types of targets.

In addition to revealing a host factor essential for disease symptom development, our study yields insight into the nature of BBC resistance mediated by *b6*. Initially characterized as recessive[57], genetic linkage mapping analysis suggests that the genetic basis of *b6*-resistance is rather complex and exhibits a quantitative inheritance. In rice, a mechanism of recessive resistance is disruption of TAL effector-dependent activation of an *OsSWEET S* gene due to polymorphism at the EBE[58]. Our data revealed that the induction of *GhSWEET10* by Avrb6 in Acb6 is compromised likely not due to the sequence polymorphism at the EBE, suggesting a novel mechanism underlying *b6*-mediated resistance. To our knowledge, this is the first report of polymorphisms outside the EBE itself that impair activation by a TAL effector. The *b6*-mediated resistance is accompanied by the HR at the site of infection, and this is triggered specifically in the presence of Avrb6 (refs 10,38). However, a dTALE-activating *GhSWEET10* only results in water-soaking, but not HR in Acb6, suggesting that *GhSWEET10* does not mediate Avb6-induced HR. This is likely that HR is due to the activation of another target by Avrb6 in Acb6, but not in Ac44E. Thus, it seems possible that *b6* comprises at least two genetic determinants, one manifesting in reduced susceptibility (lower expression of the *GhSWEET10 S* gene) and the other mediating the defence-associated HR. Perhaps both need to be present to support the full level of resistance.

Nevertheless, our findings suggest a strategy to improve BBC resistance in cotton through genome editing of TAL effector EBEs in *SWEET* genes. Genome editing has been successfully used in rice to mutate the *OsSWEET13 S* gene and generate plants resistant to *Xoo* strains that depend on TAL effector PthXo2 as a major virulence factor[27]. An advantage of this strategy is that relatively minor sequence changes could be made that will destroy the EBE while being unlikely to impinge on native promoter activity and without affecting the coding region, keeping endogenous function of the *SWEET* gene intact[51,59]. The successful knockout of all homeologs of the *Mlo* gene in hexaploid wheat in a single genome editing experiment indicates that disrupting EBEs in *SWEET* genes in cotton is likely feasible[60]. As new TAL effectors important for virulence of this pathogen are characterized in the future, this system could be used to edit the EBEs in the corresponding targets, whether *GhSWEET* genes or other BBC *S* genes yet to be discovered, and to generate plants with broad-spectrum BBC resistance via pyramiding. Not least, duplicating the Acb6 genotype of *GhSWEET10* in the Ac44E background and in other susceptible genotypes would be a powerful approach to dissect the genetic determinants of *b6*-mediated resistance, potentially by uncoupling loss of susceptibility from the HR. Such characterizations would also likely lead to new strategies for improved control of BBC.

## Methods

**Plants and bacterial strains and growth conditions.** The *Gossypium hirsutum* lines Acala44E (Ac44E), Acalab6 (Acb6) and FM706V, and *G. arboreum* line SA-1415 were grown in 3.5-inch square pots containing Metro Mix 900 soils (Sun Gro Horticulture, Agawam, MA) and tobacco *Nicotiana benthamiana* were grown in 3.5-inch square pots containing Metro Mix 366 soils in a growth chamber at 23 °C, 30% humidity and 100 µE m$^{-2}$s$^{-1}$ light with a 12-h light/12-h dark photoperiod. Two-week-old cotton plants were used for *Agrobacterium*-mediated VIGS assay, *Agrobacterium*-mediated transient protein expression assay or protoplast isolation. Plants after inoculation with *Xanthomonas citri* subsp. *malvacearum* (*Xcm*) were transferred into a growth chamber at 28 °C, 50%

humidity and 100 μE m$^{-2}$s$^{-1}$ light with a 12-h light/12-h dark photoperiod. Bacterial strains used in this study are listed in Supplementary Table 3. *E. coli* and *A. tumefaciens* strains were grown in Luria–Bertani (LB) medium with appropriate antibiotics at 37 °C or 28 °C, respectively. *Xcm* strains were grown in nutrient broth medium with appropriate antibiotics at 28 °C. The *Xcm* field strains were isolated from infected cotton fields in Texas through Koch's postulates with modifications. Water-soaked lesions were excised from infected cotton leaves using a sterile scalpel. The cut lesions were soaked and macerated in 100 μl of sterile water. The bacterial suspension was plated on nutrient agar from serial dilutions. Colonies were re-streaked and inoculated into cotton to confirm pathogenicity.

**Xcm inoculations.** The *Xcm* strains were inoculated into 4 ml of nutrient broth medium with appropriate antibiotics for overnight. For syringe infiltration, bacterial suspensions adjusted to OD$_{600}$ = 0.1 were inoculated into expanded cotyledons of 2-week-old plants or true leaves using a needleless syringe. Small holes were created with a 25G needle on the underside of the cotyledons to facilitate infiltration. For vacuum infiltration, a final bacterial suspension was made by adjusting the bacterial cultures to OD$_{600}$ = 0.01 mixed with 0.04% of Silwet L-77 surfactant solution. Plants that were silenced via VIGS 3 weeks earlier were submerged into the bacterial suspension inside a desiccator connected to a vacuum pump. The plants were vacuum infiltrated for 5 min at 76 mmHg.

**DNA isolation for Xcm whole-genome sequencing.** The *Xcm*H1005 and *Xcm*N1003 DNA used for PacBio sequencing was prepared using a protocol for total genomic DNA isolation with modifications[54]. Bacteria were cultured overnight in 30 ml glucose yeast extract media (2% glucose, 1% yeast extract) in a 250 ml flask at 28 °C on a rotary shaker at 250 rpm, collected by centrifugation at 2,580g for 10 min at 4 °C, gently resuspended and washed in 20 ml NE buffer (0.15 M NaCl, 50 mM EDTA) twice to remove the extracellular polysaccharide. Cells were then gently resuspended in 2.5 ml solution containing 50 mM Tris, pH 8.0 and 50 mM EDTA, and then 0.5 ml solution containing 25 mM Tris, pH 8.0, 10 μl ReadyLyse (Epicentre) and 50 μl RNase (10 mg ml$^{-1}$). Suspensions were mixed gently by inversion and then incubated on ice for 45 min. Following incubation on ice, 1.0 ml STEP buffer (0.5% SDS, 50 mM Tris, pH 7.5, 40 mM EDTA, protease K at 2 mg ml$^{-1}$) was added, and the lysate was mixed well by inversion and incubated at 37 °C for 1 h, mixing every 10–15 min. Next, 1.8 ml of 7.5 M ammonium acetate was added and the lysates were mixed gently before being subjected to two extractions with phenol/chloroform (10 ml) and one extraction with chloroform/isoamyl alcohol (24:1, pH 8.0, 10 ml), shaking vigorously by hand to mix and separating the aqueous and organic phases by centrifugation each time at 10,300g for 10 min at 4 °C. Following this, the aqueous phase was transferred to a 14 ml tube and DNA was precipitated by addition of two volumes of cold 95% ethanol and gentle, repeated inversion. Once solidified, the DNA was transferred to a 2 ml micro-centrifuge tube using a Pasteur pipette with the tip previously sealed and bent into a hook over a flame. Remaining liquid was then removed by centrifugation at 825g for 5 min and the pellet was washed once with 70% ethanol. The remaining liquid was removed as before and the tube left open to dry in a laminar flow hood until the edges of the pellet became glossy in appearance (10–15 min). Finally, the pellet was dissolved in 100 ml TE buffer (10 mM Tris/HCl, 1 mM EDTA, pH 8.0) overnight at 4 °C and then adjusted to a concentration of 1 μg ml$^{-1}$.

**PacBio sequencing and genome assembly.** DNA library preparation and sequencing was performed according to the manufacturer's instructions by the Genomics Core Facility of the Icahn School of Medicine at Mt. Sinai (New York, NY, USA)[54]. Separately, DNA preparation and electrophoresis to check for small plasmids in either strain that would be lost during library preparation was using standard methods[50]. Sequencing was conducted to >120 × coverage, which due to improvements in sequencing technology required only two SMRT cells per strain. In all data sets, read-length distribution showed a fat tail, with 20% of coverage after adaptor removal contained in subreads greater than or equal to 15,000 bp. The whole genomes of H1005 and N1003 were assembled using HGAP3 (ref. 61) and verified by carrying out local assemblies of reads containing *tal* genes and then comparing those assemblies to the whole-genome assembly as described in Booher et al.[54]. The assemblies conclusively match published Southern blot results using a *tal* gene probe, as well as restriction enzyme mapping and sequencing or partial sequencing of cosmid clones and subclones[10,38–40].

The initial assembly of *Xcm*H1005 showed frameshifts in *avrB104*, *avr103* and *avrb6* resulting from an extra cytosine (C) in a stretch of six Cs commonly found in the 3′ region of *tal* genes. Since *avrb6* is functional, the previously published sequence of *avrb6* shows an intact reading frame[38], and residual errors in homopolymer runs sometimes persist after consensus calling by Quiver and are a known weak point of the error-correction software (https://github.com/PacificBiosciences/GenomicConsensus/blob/master/doc/FAQ.rst), we concluded that the frameshift in *avrb6* was such an error, and we corrected it in the final assembly. Likewise, since clones containing *avrB104* and *avrB103* were functional, conferring avirulence on AcB5 and AcBIn3-containing cotton[40], we concluded that the assembly contained the same error for each of these genes, and corrected them both. No other *tal* genes in the assembly harboured this or another frameshift

within a homopolymer stretch. Note that the frameshift in *pthN'*, retained in the final assembly, is due to a deletion of a single thymine, not within a homopolymer, and *pthN'* was shown to be non-functional[10].

**Construction of avrb6 in the expression vectors.** *Avrb6* was amplified from *Xcm*H1005 plasmid DNA with primers containing restriction enzymes SpeI at the amino (N) terminus and SmaI at the carboxy (C) terminus (Supplementary Table 4) and ligated into a plant protoplast expression vector *pHBT* with a *CaMV 35S* promoter at the N terminus and a HA-epitope tag at the C terminus. To construct the *pCB302* binary vector containing *avrb6* for *Agrobacterium*-mediated transient expression assay in *N. benthamiana*, *avrb6* was released from the *pHBT* vector through digestion with SpeI and SmaI and ligated into the *pCB302* binary vector. All the clones were confirmed by sequencing.

**Transient expression assays.** Cotton protoplasts were isolated from 2-week-old cotyledons. Detached cotyledons were cut with a razor blade and digested in an enzyme solution (1.5% cellulose R10, 0.4% macerozyme R10, 0.4 M mannitol, 20 mM KCl, 20 mM MES pH 5.7) supplemented with 2% sucrose for 0.5 h under vacuum. Subsequently, the enzyme solution was incubated without vacuum at room temperature for 12 h. Protoplasts were released by filtering through a 30 μm-nylon mesh, washed with W5 solution (154 mM NaCl, 125 mM CaCl$_2$, 5 mM KCl, 2 mM MES pH 5.7) and diluted in MMG solution (0.4 M mannitol, 15 mM MgCl$_2$, 4 mM MES pH 5.7) to a density of 2 × 10$^5$ cells ml$^{-1}$ (ref. 62). A measure of 100 μl of protoplasts for each sample were collected for western blotting after expressing *avrb6-HA* for 12 h. Avrb6-HA protein was detected by immunoblotting using the α-HA antibody. *Agrobacterium tumefaciens* strain GV3101 containing *pCB302-avrb6-HA* vector or an empty vector was cultured overnight in LB medium containing appropriate antibiotics at 28 °C. Bacteria were collected by centrifugation and resuspended with infiltration buffer (10 mM MES, pH 5.7, 10 mM MgCl$_2$, 200 μM acetosyringone) at OD$_{600}$ = 0.6. Leaves of 4-week-old soil-grown *N. benthamiana* or cotyledons of 2-week-old Ac44E cotton were hand infiltrated using a needleless syringe with *Agrobacterium* cultures. Leaf samples were collected 48 h after infiltration for protein sample and immunoblot analysis. Full blots are provided in Supplementary Fig. 10.

**Cotton RNA isolation for RT–PCR and qRT–PCR analysis.** Cotton total RNA was extracted from protoplasts (500 μl protoplasts with a cell density of 2 × 10$^5$ per ml for each sample) using TRIzol reagent (Invitrogen, USA) or leaves (one cotyledon or one true leaf for each sample) using a Spectrum plant total RNA kit (Sigma-Aldrich, USA) according to the manufacturer's protocol. RNA was then treated with DNase (Invitrogen, USA) to remove genomic DNA. Purified RNA (1 μg) was subsequently reverse transcribed using a first-strand cDNA synthesis kit (Promega, USA) with oligo(dT) as a primer. PCR amplification was performed for 30 cycles of 15 s at 98 °C, 30 s at 55 °C and 45 s at 72 °C with primers listed in Supplementary Table 4. GhACTIN was used as an internal control for RT–PCR. qRT–PCR analysis was performed using iTaq SYBR green Supermix (Bio-Rad, USA) with a Bio-Rad CFX384 Real-Time PCR System. Expression of each gene was normalized to the expression of GhUBQ1.

**RNA sequencing and data analysis.** A measure of 500 μl of cotton protoplasts (2 × 10$^5$ per ml) were transfected with 35S::avrb6-HA or an empty vector control. Protoplasts were collected 12 h after transfection and RNA was extracted by the TRIzol reagent. Three independent biological repeats were performed for the RNA-Seq analysis. For each repeat, equal amount of RNA from two biological replicates was pooled for RNA-Seq library construction. RNA-Seq libraries were prepared using Illumina TruSeqTM RNA Sample Preparation Kit following the manufacturer's instruction, and sequenced on an Illumina HiSeq 2000 platform with 125-nucleotide pair-end reads at Texas AgriLife Genomics and Bioinformatics Service (College Station, TX). RNA-Seq read processing, alignment to the *G. raimondii* genome and differential gene expression analysis were performed as described in Li et al.[63]. Genes with expression fold change ≥ 2 and *P* value < 0.05 were considered as significantly different between samples with and without Avrb6.

**EBE prediction.** All TAL effector EBE predictions were conducted using the TALE-NT 2.0 Target Finder tool[64] to search the promoter regions defined in the *G. raimondii* genome as the 1,000 bp upstream of the transcriptional start site plus the 5′ UTR, if annotated. We identified promoter sequences for *G. raimondii* using genome assembly v2.0 and annotation v2.1 (ref. 3). Binding sites passing the standard target finder score ratio cutoff of 3.0 were ranked based on both target finder output and genomic context using the machine-learning classifier presented by Cernadas et al.[35] and updated by Wilkins et al.[65].

**Construction of VIGS vector and Agrobacterium-mediated VIGS.** GhSWEET10 was amplified by PCR from cDNA of *G. hirsutum* with primers described in Supplementary Table 4, and inserted into the pYL156 (pTRV-RNA2) vector with restriction enzymes EcoRI and KpnI, resulting in pYL156-GhSWEET10. The *Agrobacterium tumefaciens* strain GV3101 carrying the plasmids pTRV-RNA1 and pYL156 (pTRV-RNA2) or pYL156-GhSWEET10 was collected and resuspended

with infiltration buffer (10 mM MES, pH 5.7, 10 mM MgCl$_2$, 200 μM acetosyringone) at OD$_{600}$ = 0.75. The culture suspension of pTRV-RNA1 was mixed with pYL156 or pYL156-GhSWEET10 at a 1:1 ratio and hand infiltrated using needleless syringes into two fully expanded cotyledons of 2-week-old plants[66].

**GhSWEET10D transporter analysis in HEK293T cells.** HEK293T cells were co-transfected with FRET sucrose sensor FLIPsuc90μδ3A and GhSWEET10D in six-well plates and perfusion experiments were performed[48]. After 3.5 min buffer perfusion, cells were perfused with 25 mM sucrose for 4 min followed by 2.5 min buffer perfusion. Images were captured at the following settings: exposure time 800 ms, gain 3, binning 2, time interval 10 s. Fluorescence intensity of selected individual cells was recorded. The quantitative ratio of eCFP and Aphrodite emission at excitation of eCFP was calculated after the background was subtracted. All of ratios were normalized to the initial ratio.

**Yeast mutant EBY4000 complementation growth assay.** The yeast strain EBY4000 was transformed with the different plasmids following the standard protocol. Spotting assay with 10-fold serial dilutions was performed. Yeast cells were diluted to OD$_{600}$ = 0.22 in water after cells grew to OD$_{600}$ = 0.6, followed by a twofold dilution. Then, 10-fold serial dilutions were plated on synthetic medium containing either 2% maltose (as a control), 2% fructose or 2% glucose. Growth was documented by Gel-doc system after 2–4 days at 30 °C.

**Transactivation assays.** The reporter constructs, pGhSWEET10D::LUC, pGh067700::LUC, pGhKBS1::LUC, pGhMDR1::LUC, pGhHLH1::LUC and pGaSWEET10::LUC, were constructed by amplifying the promoters of candidate genes (∼800 bp upstream of translational start site) from genomic DNA of Ac44E, Acb6 or G. arboreum and were fused to a luciferase gene at the C terminus. Protoplasts were transfected with the effector and reporter construct and expressed at room temperature for 12 h. Protoplasts were then collected and lysed and the luciferase activity was detected by Glomax Multi-Detection System (Promega, USA) with the luciferase assay substrate (Promega, USA). UBQ10-GUS was included in all samples as the internal transfection control and the GUS activity was analysed with a Multilabel Plate Reader (Perkin-Elmer, USA). The pGhSWEET10D-mEBE::LUC construct was generated by the site-directed mutagenesis PCR using plasmid DNA of pGhSWEET10D::LUC as the template and primers are listed in Supplementary Table 4.

**Construction of dTALEs.** The dTALEs were constructed using golden gate cloning using a complete plasmid kit available through AddGene[67]. Four types of repeats encoding the RVDs NI, NN, NG and HD that correspond to the respective nucleotide A, G, T and C were used to assemble the repeat domains of the artificial dTALEs. The dTALEs were transformed into Xcm stains via bacterial tri-parental mating by using a helper plasmid, pRK2073 (ref. 38). In addition, the modifier plasmid pUFR054 was used to increase the efficiency of transferring the plasmids from E. coli DH5α to Xcm strains. The bacterial mating was carried out by mixing the donor, recipient, modifier and helper strains in a 1:1:1:1 ratio. The mixed culture was plated onto nutrient agar medium with appropriate antibiotics. Colonies were confirmed via Sanger sequencing of extracted plasmid DNA.

**Measurement of sucrose concentration.** Leaves from 4-week-old N. benthamiana plants were infiltrated with Agrobacterium carrying 35S::GhSWEET10D-HA or an empty vector control at OD$_{600}$ = 1.0. Leaves were detached 3 dpi and fully vacuum infiltrated with sterile Millipore water. The leaves were blot dried, cut into small pieces with a razor blade and placed in a nylon mesh strainer at the top of a 50 ml centrifuge tube. The apoplastic solution was collected through centrifugation at 3,220g for 20 min at 4 °C. The concentration of sucrose was determined by using a sucrose assay kit (Sigma-Aldrich).

**Data availability.** The XcmH1005 genome assembly has been deposited in GenBank under accessions CP013004 (chromosome) and CP013005 (pXcmH) and the XcmN1003 assembly under accessions CP013006 (chromosome) and CP013007 (pXcmN). Raw data as bas.h5 and bax.h5 files and associated metadata are available in the NCBI sequence read archive (SRA) under SRP065335 for XcmH and SRP065336 for XcmN. The RNA-Seq data were deposited in the NCBI SRA under accession number SRP100553. The authors declare that all other data supporting the findings of this study are included within the manuscript and its supplementary files or are available from the corresponding authors on request.

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

## Acknowledgements

We thank Dr Dean Gabriel for various *Xcm* strains, Dr Margaret Essenberg and Jared Harris for the seeds, Stephanie Yamashita and Tri Tran for excellent technical assistance and Karla Vogel for constructing pKEB1. This work was supported by the USDA NIFA (2012-67013-19433) to P.H. and L.S., the Cotton Inc (15-174) to L.S., Texas AgriLife Research Cotton Improvement Program to L.S., P.H. and T.W., the US National Science Foundation (Plant Genome Research Program Award IOS 1238189) to A.J.B., the Texas A&M University Graduate Diversity Fellowship and USDA NIFA AFRI Pre-doctoral Fellowship (Award No. 2017-67011-26060) to K.L.C., Graduate Research Fellowship Award (Grant No. DGE-1144153) to K.E.W.

## Author contributions

K.L.C., F.M., K.E.W., P.H., A.B., L.S. conceived the study and designed the experiments; K.C., F.M., K.W., F.L., P.W., N.J.B., S.C.D.C., L.-Q.C., H.Z., X.G., Y.Z., and Z.F. performed the experiments and/or data analysis; and K.C., P.H., A.J.B. and L.S. drafted the manuscript. The manuscript and figures were reviewed and revised by all the authors.

## Additional information

**Competing interests:** The authors declare no competing financial interests.

