## [Peer Review File · Nature Communications]

Reviewers' comments:

Reviewer #1 (Remarks to the Author):

The manuscript of Cox and coauthors is a very thorough study of the virulence function of a transcription activator-like effector (TALE) from cotton-pathogenic bacteria. *Xanthomonas citri* subsp. *malvacearum* (Xcm) are important pathogens of cotton and known to contain multiple TALEs per bacterial strain. TALEs are translocated from the bacteria into plant cells to induce expression of specific target genes. Although some TALEs, like AvrB6, have been shown to contribute to virulence of Xcm, no target induced gene in cotton has been identified, previously.

The authors determined the genome sequences including the TALE repertoires of two Xcm strains. Using TALE-target prediction and transcriptome profiling, they obtained a list of candidate targets for AvrB6 and verified that GhSWEET10 is a direct target for transcriptional induction by AvrB6. The authors confirmed that GhSWEET10 encodes a functional sucrose transporter, similar to related proteins from other plants. Several Xcm strains apparently induce the expression of an alternative SWEET gene from cotton which supports the notion that different related members of this class of transporters can possibly support virulence of Xcm on cotton, an observation that has been described from rice, earlier. Different SWEET sugar transporters of a related sub-class are the best known examples for TALE targets and are well known to support *Xanthomonas* proliferation in rice and cassava, but have not been known to be important for bacterial blight of cotton.

This study is technically very well done and applies many experimental approaches to identify and characterize the virulence activity of AvrB6. In principle the authors have followed the line of experiments that have successfully been used several times before and are standard for identifying and characterizing TALE targets. SWEET transporters are well known TALE targets and are well known to support *Xanthomonas* infections, but the finding that this is also the case for infection of cotton is significant. Apparently, SWEET-mediated sugar transport is not only important for *Xanthomonas* infections of rice and cassava, but also of cotton.

Overall, this is a solid study of high interest but predominantly for the scientific community in the plant pathogen field. It has possible implications for future approaches to generate resistant cotton lines, but these are not necessarily novel (see specific points below).

Specific points:

- the introduction can be shortened (e.g. lines 76-80 and others), while other aspects need to be more precise.
- lines 94-95: the authors state that the sequence of RVDs defines the EBE. To my knowledge, TALEs exhibit an additional DNA-specificity, which is an initial "T" preceding the target sequence and which is mediated by the N-terminal domain of the TALE. Is this not part of the EBE? Please specify.
- lines 96-97: to be correct, the first identified TALE target was identified in rice and published by Yang et al (2006), reference [19]
- line 103: to be consistent "...Os8N3/Xa13..." might be written as "...Os8N3/Xa13/OsSWEET11..."
- line 114: the authors might consider to add Xa23 as a known target (and R gene) to the list.
- lines 142-143: the phrasing "...and provide a new strategy for developing cotton varieties carrying broad-spectrum resistance to this disease." remains obscure to the reader at this point. I am in general not convinced about this (see further below).
- line 171: please indicate to the unexperienced reader whether the cotton genotype Ac44E is diploid or tetraploid.
- lines 175-179: I am not in favor of using protoplasts expressing a single TALE to identify target genes via RNAseq. Overexpression of the TALE will result in plant genes being induced which will likely not be induced in a natural infection. This point is not critical for the present manuscript, because the

major target (GhSWEET10) is well confirmed in additional experiments, but the authors should take care for future experiments.

- lines 204-210: listings of gene names in the main body of the text makes this part difficult to read. They might be better placed into a table.

- lines 231-233: there are no controls that indicate whether CHX is suppressing protein synthesis in this particular experimental setup.

- lines 255-269: dTALE4 is not functional for reasons which remain unknown and which might well be a simple technical issue, e.g. that the basal expression level of the GhKBS1 gene is already very high (as it is visible in Figure 4C) and an additional TALE-mediated expression is not significant. Because this experiment can not conclusively be interpreted, I suggest to remove the data concerning dTALE4 from Figure 4 and from the text.

- line 287, I believe the last word in the line should be "A-subgenome" (not "D-subgenome")

- Fig. 5D is not convincing. Why are the symptoms of the complemented mutant strains (H1005Davr6+dTALE1-1/2) on either plant so much weaker than the wildtype strain? Why does the wildtype strain cause so strong symptoms on the VIGS silenced plant (stronger than the avrb6 mutant and even stronger than the complemented mutant on non-silenced plants)? The quality of the pictures is also very low. In my opinion, Figure 5, panels A-C are sufficient to demonstrate that silencing of GhSWEET10 interferes with disease development, therefore, I suggest to remove panel D.

- lines 361-362, 376: are the changes in the GhSWEET10 promoter sequence that are found in the cotton line Ac6 present in the A and D genome?

- I am not convinced that the changes to the GhSWEET14 promoter are the cause for the recessive b6-resistance. The authors' infection studies of the cotton line Ac6 with the multiple TALE mutant HM2.2S(placZ::avr6) lead to browning of the infection area (Fig. S7A). This phenotype is interpreted by the authors as an HR (line 362), although an HR is typically a dominant reaction. Is this HR-like phenotype also visible, if the wildtype strain H1005 is used? What about an Avr6 mutant? If the browning phenotype can be directly related to the presence of avrb6 it would be an indication that this TALE is involved in this reaction, but so far, it can also be caused by any other combination of factors. In addition, if an HR (programmed cell death) is triggered during infection of Ac6 this might indirectly affect GhSWEET10 induction (Fig. 7B) and the promoter difference might not be the cause.

Furthermore, it is difficult to understand, why the relatively low expression of GhSWEET10 in Ac6 would cause an HR whereas an inability to induce the gene in another cotton line (in the VIGS silenced lines or by strains not delivering Avr6) does not. How does a strain delivering dTALE2 instead of Avr6 perform on the cotton Ac6 line (the dTALE1 EBE in Ac6 carries a critical mutation and might not respond to this TALE)? dTALE2 is binding slightly downstream of the Avr6 EBE and it would be interesting to know whether or not this TALE can induce SWEET10. In any case, this promoter mutation will not provide a broad spectrum resistance (as mentioned above), because other Xcm strains apparently trigger induction of other SWEET members in cotton (Figure 7D). Taken together, the authors should reword their conclusions of the experiments with Ac6 and display different possible scenarios to explain these results.

- lines 384-399: please refer to Figure 7D somewhere in this paragraph.

- the discussion is not very inspiring, in particular the section from lines 407-417 is a repetition of the summary.

- lines 435-440: the model that an individual TALE (Tal3-XcmH1005 / Tal8a-XcmN1003) from the Xcm strains can induce multiple SWEET genes is very speculative and likely not the case for several reasons. (1) The H1005Davr6 mutant does not induce GhSWEET10 and is not virulent, which indicates that there is no other TALE in this strain able to induce this gene (although there is an EBE predicted) or any other SWEET from this clade. (2) The same is apparent for GhSWEET11, 12a, 14a. (3) A quick manual comparison of RVDs and EBEs revealed at least four non-matching positions within the EBEs including the highly sensitive early repeats (I did not check all EBEs, but only the most probable ones). Current understanding is that this amount of non-matching combinations will completely block the activity of a TALE with only 16 repeats. (4) As a comparison: Avr6 shows no

activity on an EBE with only three non-matching RVD-base combinations (Figure 3G). Therefore, I recommend to delete Table S1 and this paragraph from the discussion.

- line 469-471: to acknowledge genome editing of EBEs with the aim to block bacterial virulence, two additional references should be added here: Li et al., 2012 Nat. Biotechnol.; Blanvillain-Baufume et al., 2016 Plant Biotech. J.

- I have a slight reservation to the experiment in Figure 5C. The data show that GhSWEET10 is much less expressed in the VIGS-silenced plants. Because the RNA was sampled three weeks after infection the expression difference is the result of lower induction by the TALE and less growth of the bacteria. How much of the effect can be attributed to each factor remains unknown. An alternative would be to inoculate the bacteria with higher cellular density and sample the RNA for RT-PCR two days after inoculation. The differential ability to grow would in this case be negligible and the difference in expression could be referred to the TALE activity in vivo. This experiment might be a lot of work for a rather minor point, so I suggest that the authors just add a sentence or two in the text to acknowledge it.

- Legend to Figure 1: "...from outside to inside..." should be "...from inside to outside..."

- Figure 1D: two TALEs apparently have only one repeat. Please indicate whether there is a frameshift that prohibits translation of more repeats (but the DNA coding sequence is present in another frame) or whether one repeat is present in a fully functional TALE (with correct N- and C-terminal regions)

- Figure 2 is trivial and should be moved to the supplemental section

- The legend to Figure 3 is much too long and needs to be shortened significantly. Legends to Figures 4, 5, 6, and 7 are also too long.

- Figure 4 E+F: please indicate that the sequences correspond to "EBEs"

- References: please check formatting of titles, journal abbreviations, page numbers, etc., because there are several inconsistencies.

Reviewer #2 (Remarks to the Author):

This is a well-performed and interesting study, which was a joy to read - congratulations to the authors! This study builds on accumulating evidence that SWEET transporters – and maybe released sugars acting as carbon sources – are important for many infection diseases of plants. Much work has been done with bacteria of the genus *Xanthomonas*, which typically have a narrow host range but collectively infect a plethora of plant species. Following infection by xanthomonads SWEETs have been shown to be induced in rice and cassava leaves, where they contribute to the virulence of the pathogen, but also in citrus and pepper plants, where the role in pathogenicity is less clear. Here, state-of-the-art approaches and techniques were used to demonstrate a key role for SWEETs in the *Xanthomonas*-cotton interaction. This knowledge will be instrumental to generate and breed cotton cultivars that are resistant to attack by *Xanthomonas citri* pv. *malvacearum* (XCM), the causal agent of cotton bacterial blight. The key discoveries of the manuscript are:

(i) Strains of XCM encode complex repertoires of TAL effectors that may induce the expression of numerous plant genes.

(ii) RNA-seq, virus-induced gene silencing and bioinformatics experiments revealed a candidate susceptibility gene, SWEET10, which is targeted by the TAL effector Avr_{b6} and which is responsible for causing water-soaking in cotton.

(iii) *Gossypium hirsutum* SWEET10D is a member of a large gene family in cotton and encodes a functional sucrose transporter.

(iv) Resistance mediated by the *b6* resistance locus involves a compromised responsiveness of the GhSWEET10 gene to Avr_{b6}-mediated gene induction likely due to promoter polymorphisms around the TAL effector binding site but may also involve another genetic determinant that mediates a defense-associated hypersensitive response.

(v) Emergent XCV isolates in the United States of America were unable to induce GhSWEET10 but highly induced GhSWEET14, another clade-III SWEET gene; a finding that is reminiscent of previous observations in rice.

Specific comments:

(A) Unquestionable, induction of SWEETs appear to be important for disease. Nevertheless this reviewer would have appreciated a cautionary note that SWEET induction might become bypassed by certain isolates, as has been shown recently for rice-pathogenic xanthomonads (Blanvillain-Baufumé et al, *Plant Biotechnol. J.*, in press).

Correctness of assembly of TAL effector genes from PacBio reads should be supported by Southern blot experiments, ideally separately for plasmid DNA and chromosomal DNA.

A comparison of the genomic context – synteny – of the TAL effector gene loci would be interesting. It appears that both strains contain different numbers of TAL effector gene clusters, and orthologs appear to be encoded at different chromosomal and/or plasmidic locations (e.g. chromosomal Tal3-XcmH1005 versus plasmidic Tal8aXcmN1003). A scheme showing where the different TAL effector genes are encoded could be included in Figure 1.

The nomenclature of TAL effector genes is confusing. Apparently, Tal3 in one strain is an ortholog of Tal8a in the other strain, in which then Tal3 corresponds to Tal4 in the first strain. This reviewer strongly suggests adapting a universal nomenclature, which will help in the future to keep track of the various TAL effector genes in different isolates. Such a nomenclature has been, for example, suggested by Grau et al. 2016 (*Sci. Rep.* 6: 21077).

Minor points:

Figure 1: It's hard if not impossible to distinguish protein-coding genes (blue) from rRNA genes (lavender), at least at the current resolution. Anyway, one may ask if these plots provide any useful information or if they merely fulfill esthetic purposes? By the way, only chromosomes are shown in the figure, not the whole genomes.

Line 253: The data presented merely confirm one EBE (SWEET10D) as being required for Avr6-induced expression but not for the other two genes.

Line 338: Here, it's more common to use the term "query" instead of "bait".

Reviewer #3 (Remarks to the Author):

By comparing transcriptome data and TAL EBE predicted by computational analysis, the authors found that GhSWEET10 is one of the targets of Avr6, which is a TAL effector in XcmH1005. Throughout the experiments described in this manuscript, it is nicely demonstrated that induction of GhSWEET10 expression by Avr6 is required for normal BBC symptom development upon infection by Xcm.

It has been suggested that sucrose transport mediated by SWEET proteins plays a role in providing the bacteria with a carbon source. Based on this, the authors determined the sugar transport activity of GhSWEET10. Using a previously established heterologous system with the human HTK293T cells, the authors confirmed that GhSWEET10 has an ability to facilitate sucrose uptake into the cells. However, according to the hypothesis mentioned above, the role of SWEET proteins in the pathogen infection should be related to its sugar export activity. Thus, the authors need to determine whether GhSWEET10 has a sugar transport activity at least in a heterologous system such as *Xenopus* oocytes. Or an in planta evidence could be supported by measuring sugar concentrations in apoplast after overexpressing GhSWEET10 as described in Zhou et al. (*Plant J*, 2015, 82, 632-643). In addition, it has been shown that SWEET proteins transport glucose and fructose in addition to sucrose. So I would suggest the authors to determine whether GhSWEET10 also transports these sugars. It was recently reported that *Arabidopsis* SWEET proteins in the clade III transport the plant hormone gibberellin (Kanno et al., *Nat Commun*, 2016, 7, 13245). It would be helpful to readers to cite this report in the

present manuscript.

Finally, the authors showed that different GhSWEET proteins in the clade III are induced depending on different Xcm field isolates. This suggests that clade III SWEETs are generally important for Xcm strains to cause BBC and that some field isolates evolved new TAL effectors to target new SWEET genes. This is an interesting hypothesis. However, I think more careful discussion (for example the presence of unidentified substrate(s) for SWEETs) would be required because it is not clear that all the GhSWEETs in the clade III have the same function (at least biochemically).

Reviewer #1 comments:

-The manuscript of Cox and coauthors is a very thorough study of the virulence function of a transcription activator-like effector (TALE) from cotton-pathogenic bacteria. *Xanthomonas citri* subsp. *malvacearum* (Xcm) are important pathogens of cotton and known to contain multiple TALEs per bacterial strain. TALEs are translocated from the bacteria into plant cells to induce expression of specific target genes. Although some TALEs, like Avr6, have been shown to contribute to virulence of Xcm, no target induced gene in cotton has been identified, previously. The authors determined the genome sequences including the TALE repertoires of two Xcm strains. Using TALE-target prediction and transcriptome profiling, they obtained a list of candidate targets for Avr6 and verified that GhSWEET10 is a direct target for transcriptional induction by Avr6. The authors confirmed that GhSWEET10 encodes a functional sucrose transporter, similar to related proteins from other plants. Several Xcm strains apparently induce the expression of an alternative SWEET gene from cotton which supports the notion that different related members of this class of transporters can possibly support virulence of Xcm on cotton, an observation that has been described from rice, earlier. Different SWEET sugar transporters of a related sub-class are the best known examples for TALE targets and are well known to support *Xanthomonas* proliferation in rice and cassava, but have not been known to be important for bacterial blight of cotton. This study is technically very well done and applies many experimental approaches to identify and characterize the virulence activity of Avr6. In principle the authors have followed the line of experiments that have successfully been used several times before and are standard for identifying and characterizing TALE targets. SWEET transporters are well known TALE targets and are well known to support *Xanthomonas* infections, but the finding that this is also the case for infection of cotton is significant. Apparently, SWEET-mediated sugar transport is not only important for *Xanthomonas* infections of rice and cassava, but also of cotton. Overall, this is a solid study of high interest but predominantly for the scientific community in the plant pathogen field. It has possible implications for future approaches to generate resistant cotton lines, but these are not necessarily novel (see specific points below).

We appreciate the reviewer's comments on our manuscript.

-The introduction can be shortened (e.g. lines 76-80 and others), while other aspects need to be more precise.

We thank the reviewer for the suggestion and we have modified the introduction and other parts as suggested.

-Lines 94-95: the authors state that the sequence of RVDs defines the EBE. To my knowledge, TALEs exhibit an additional DNA-specificity, which is an initial "T" preceding the target sequence and which is mediated by the N-terminal domain of the TALE. Is this not part of the EBE? Please specify.

We thank the reviewer for pointing out this oversight. Convention (and our habit) is to treat the T0 not as part of the EBE per se (in at least one case it is a C and for *Ralstonia* TALEs it is a G).

-Lines 96-97: to be correct, the first identified TALE target was identified in rice and published by Yang et al (2006), reference [19]

We have modified the text accordingly.

-Line 103: to be consistent "...Os8N3/Xa13..." might be written as ...Os8N3/Xa13/OsSWEET11..."

We have revised this phrase in the manuscript as suggested.

-Line 114: the authors might consider to add Xa23 as a known target (and R gene) to the list.

We have added Xa23 to the list as suggested.

-Lines 142-143: the phrasing "...and provide a new strategy for developing cotton varieties carrying broad-spectrum resistance to this disease." remains obscure to the reader at this point. I am in general not convinced about this (see further below).

We have deleted this sentence.

-Line 171: please indicate to the unexperienced reader whether the cotton genotype Ac44E is diploid or tetraploid.

We have indicated that Ac44E is a tetraploid as suggested.

-Lines 175-179: I am not in favor of using protoplasts expressing a single TALE to identify target genes via RNAseq. Overexpression of the TALE will result in plant genes being induced which will likely not be induced in a natural infection. This point is not critical for the present manuscript, because the major target (GhSWEET10) is well confirmed in additional experiments, but the authors should take care for future experiments.

We thank the reviewer for the suggestion. We have validated the data derived from protoplasts using the leaf-bacterium infection assays. We will definitely consider this reviewer's suggestion for future experiments.

-Lines 204-210: listings of gene names in the main body of the text makes this part difficult to read. They might be better placed into a table.

Please see Table 2 for the list of the gene names.

-Lines 231-233: there are no controls that indicate whether CHX is suppressing protein synthesis in this particular experimental setup.

As suggested, we have added the proper controls to show that CHX suppressed protein synthesis (Fig. S2).

-Lines 255-269: dTALE4 is not functional for reasons which remain unknown and which might well be a simple technical issue, e.g. that the basal expression level of the GhKBS1 gene is already very high (as it is visible in Figure 4C) and an additional TALE-mediated expression is not significant. Because this experiment can not conclusively be interpreted, I suggest to remove the data concerning dTALE4 from Figure 4 and from the text.

We concur that the experiments with dTALE4 are not conclusive with unclear reasons and we have clarified in the text. At this point, we would like to keep this data in the manuscript in order to avoid confusion why we didn't further pursue GhKBS1 as a target.

-Line 287, I believe the last word in the line should be "A-subgenome" (not "D-subgenome")

We have changed it to A-subgenome.

-Fig. 5D is not convincing. Why are the symptoms of the complemented mutant strains (H1005Davr_{b6}+dTALE1-1/2) on either plant so much weaker than the wildtype strain? Why does the wildtype strain cause so strong symptoms on the VIGS silenced plant (stronger than the avr_{b6} mutant and even stronger than the complemented mutant on non-silenced plants)? The quality of the pictures is also very low. In my opinion, Figure 5, panels A-C are sufficient to demonstrate that silencing of GhSWEET10 interferes with disease development, therefore, I suggest to remove panel D.

We have removed Figure 5D from the figures and text as suggested.

-Lines 361-362, 376: are the changes in the GhSWEET10 promoter sequence that are found in the cotton line Ac_{b6} present in the A and D genome?

We have added the promoter sequences of *GhSWEET10A* and *GhSWEET10D* from Ac_{b6} and Ac44E (see Figure S4B, S9C).

-I am not convinced that the changes to the GhSWEET14 promoter are the cause for the recessive b₆-resistance. The authors' infection studies of the cotton line Ac_{b6} with the multiple TALE mutant HM2.2S(placZ::avr_{b6}) lead to browning of the infection area (Fig. S7A). This phenotype is interpreted by the authors as an HR (line 362), although an HR is typically a dominant reaction. Is this HR-like phenotype also visible, if the wildtype strain H1005 is used? What about an Avr_{b6} mutant? If the browning phenotype can be directly related to the presence of avr_{b6} it would be an indication that this TALE is involved in this reaction, but so far, it can also be caused by any other combination of factors. In addition, if an HR (programmed cell death) is triggered during infection of Ac_{b6} this might indirectly affect GhSWEET10 induction (Fig. 7B) and the promoter difference might not be the cause. Furthermore, it is difficult to understand, why the relativ low expression of GhSWEET10 in Ac_{b6} would cause an HR whereas an inability to induce the gene in another cotton line (in the VIGS silenced lines or by strains not delivering Avr_{b6}) does not. How does a strain delivering dTALE2 instead of Avr_{b6} perform on the cotton Ac_{b6} line

(the dTALE1 EBE in Acb6 carries a critical mutation and might not respond to this TALE)? dTALE2 is binding slightly downstream of the Avr6 EBE and it would be interesting to know whether or not this TALE can induce SWEET10. In any case, this promoter mutation will not provide a broad spectrum resistance (as mentioned above), because other Xcm strains apparently trigger induction of other SWEET members in cotton (Figure 7D). Taken together, the authors should reword their conclusions of the experiments with Acb6 and display different possible scenarios to explain these results.

We thank the reviewer for these comments. As suggested by this reviewer, we performed HR assay with WT strain XcmH1005 and its avr6 deletion mutant XcmH1407 in cotton Acb6. As shown in Figure S9B, XcmH1005, but not Xcm1407, caused an HR in Acb6. Similarly, HM2.2S carry avr6, not HM2.2S carrying an empty vector, caused an HR. Thus, this HR is caused by Avr6 in Acb6. However, we cannot conclude that the HR is associated with the reduced GhSWEET10 expression. As this reviewer pointed out, VIGS of GhSWEET10 did not cause an HR. We hypothesize that Avr6 may have at least two targets. One target is GhSWEET10, which is associated with susceptibility. Another target might be associated with HR. Thus, the resistance and HR induced by avr6 in Acb6 might be controlled by two different genes. Consistent with this hypothesis, inoculation of the Avr6 mutant carrying dTALE2 caused water-soaking, not HR in Acb6 (see Fig. 9C), which was an experiment suggested by this reviewer. We are sorry that we didn't make this clear in our original manuscript. We have commented on them in the discussion to clarify the model for b6-mediated resistance. Regarding the promoter mutation and broad spectrum resistance, we occur with this reviewer and have clarified in the text.

-Lines 384-399: please refer to Figure 7D somewhere in this paragraph.

We have referred to Figure 7D in the paragraph as suggested.

-The discussion is not very inspiring, in particular the section from lines 407-417 is a repetition of the summary.

We thank the reviewer for the suggestion of the discussion section. We have modified it to try to make it more appealing and inspiring to readers.

-Lines 435-440: the model that an individual TALE (Tal3-XcmH1005 / Tal8a-XcmN1003) from the Xcm strains can induce multiple SWEET genes is very speculative and likely not the case for several reasons. (1) The H1005Davr6 mutant does not induce GhSWEET10 and is not virulent, which indicates that there is no other TALE in this strain able to induce this gene (although there is an EBE predicted) or any other SWEET from this clade. (2) The same is apparent for GhSWEET11, 12a, 14a. (3) A quick manual comparison of RVDs and EBEs revealed at least four non-matching positions within the EBEs including the highly sensitive early repeats (I did not check all EBEs, but only the most probable ones). Current understanding is that this amount of non-matching combinations will completely block the activity of a TALE with only 16 repeats. (4) As a comparison: Avr6 shows no activity on an EBE with only three non-matching RVD-base combinations (Figure 3G). Therefore, I recommend to delete Table S1 and this paragraph from the discussion.

We thank the reviewer for the comment and have deleted Table S1 and this paragraph from the discussion.

-Line 469-471: to acknowledge genome editing of EBEs with the aim to block bacterial virulence, two additional references should be added here: Li et al., 2012 Nat. Biotechnol.; Blanvillain-Baufume et al., 2016 Plant Biotech. J.

We have added these two additional references as suggested.

-I have a slight reservation to the experiment in Figure 5C. The data show that GhSWEET10 is much less expressed in the VIGS-silenced plants. Because the RNA was sampled three weeks after infection the expression difference is the result of lower induction by the TALE and less growth of the bacteria. How much of the effect can be attributed to each factor remains unknown. An alternative would be to inoculate the bacteria with higher cellular density and sample the RNA for RT-PCR two days after inoculation. The differential ability to grow would in this case be negligible and the difference in expression could be referred to the TALE activity in vivo. This experiment might be a lot of work for a rather minor point, so I suggest that the authors just add a sentence or two in the text to acknowledge it.

We apologize for being unclear in the figure legend. The RT-PCR was done before *Xcm* infection. It was three weeks after *Agrobacterium* inoculation for VIGS assay. Thus, there is no complication of different *Xcm* growth rate in this experiment. As this reviewer suggested, we inoculated bacteria with high density (0.01 OD). We have clarified this in the legend for Figure 5C.

-Legend to Figure 1: "...from outside to inside..." should be "...from inside to outside..."

We have generated a new Figure 1 as suggested by Reviewer 2. Therefore, this was deleted.

-Figure 1D: two TALEs apparently have only one repeat. Please indicate whether there is a frameshift that prohibits translation of more repeats (but the DNA coding sequence is present in another frame) or whether one repeat is present in a fully functional TALE (with correct N- and C-terminal regions)

We have modified the manuscript as suggested that the C-terminal region gets cut off by an IS element in *Tal1_{XcmN1003}*, and there is an intergrase insertion in repeat region that causes a frameshift in *Tal2_{XcmN1003}*. We have also added a Supplementary Table (S3) with a complete description of all the *XcmH1005* and *XcmN1003* TAL effector genes.

-Figure 2 is trivial and should be moved to the supplemental section

We thank the reviewer for the suggestion. If space allows, we would like to keep these data in the Figure section. This figure presents the overall strategy used in this manuscript, which will be very helpful for the readers.

-The legend to Figure 3 is much too long and needs to be shortened significantly. Legends to Figures 4, 5, 6, and 7 are also too long.

We have shortened the legends to Figures 3, 4, 5, 6, and 7 as suggested and made sure they were less than the journal's 350-word limit.

-Figure 4 E+F: please indicate that the sequences correspond to "EBEs"

We have put the sequences corresponding to EBEs on the top of Figures.

-References: please check formatting of titles, journal abbreviations, page numbers, etc., because there are several inconsistencies.

We have carefully checked the references again and adjusted the formatting to make it consistent.

Reviewer 2 comments:

This is a well-performed and interesting study, which was a joy to read - congratulations to the authors! This study builds on accumulating evidence that SWEET transporters – and maybe released sugars acting as carbon sources – are important for many infection diseases of plants. Much work has been done with bacteria of the genus *Xanthomonas*, which typically have a narrow host range but collectively infect a plethora of plant species. Following infection by xanthomonads SWEETs have been shown to be induced in rice and cassava leaves, where they contribute to the virulence of the pathogen, but also in citrus and pepper plants, where the role in pathogenicity is less clear. Here, state-of-the-art approaches and techniques were used to demonstrate a key role for SWEETs in the *Xanthomonas*-cotton interaction. This knowledge will be instrumental to generate and breed cotton cultivars that are resistant to attack by *Xanthomonas citri* pv. *malvacearum* (XCM), the causal agent of cotton bacterial blight. The key discoveries of the manuscript are:

- (i) Strains of XCM encode complex repertoires of TAL effectors that may induce the expression of numerous plant genes.
- (ii) RNA-seq, virus-induced gene silencing and bioinformatics experiments revealed a candidate susceptibility gene, SWEET10, which is targeted by the TAL effector Avrb6 and which is responsible for causing water-soaking in cotton.
- (iii) *Gossypium hirsutum* SWEET10D is a member of a large gene family in cotton and encodes a functional sucrose transporter.
- (iv) Resistance mediated by the b6 resistance locus involves a compromised responsiveness of the GhSWEET10 gene to Avrb6-mediated gene induction likely due to promoter polymorphisms around the TAL effector binding site but may also involve another genetic determinant that mediates a defense-associated hypersensitive response.
- (v) Emergent XCV isolates in the United States of America were unable to induce GhSWEET10 but highly induced GhSWEET14, another clade-III SWEET gene; a finding that is reminiscent of previous observations in rice.

We thank the reviewer's comments on our manuscript.

-Unquestionable, induction of SWEETs appear to be important for disease. Nevertheless this reviewer would have appreciated a cautionary note that SWEET induction might become bypassed by certain isolates, as has been shown recently for rice-pathogenic xanthomonads (Blanvillain-Baufumé et al, Plant Biotechnol. J., in press).

We thank the reviewer for this suggestion, and have addressed and cited this work in the discussion section.

-Correctness of assembly of TAL effector genes from PacBio reads should be supported by Southern blot experiments, ideally separately for plasmid DNA and chromosomal DNA.

We share this reviewer's concern. Full concordance with previously published blots for these strains and data from *tal* gene sub-cloning and characterization, which are cited in the manuscript, provide confirmation of our assembly. Although using a standard assembly approach or less than sufficient coverage might result in an erroneous assembly (Booher et al., 2015), using the PBX toolkit and a consensus approach to assembly, as we did here, with sufficient coverage, accurately and completely assembles even the most complex *Xanthomonas* genomes, including the *tal* genes. Thus, we have full confidence in our assemblies.

-A comparison of the genomic context – synteny – of the TAL effector gene loci would be interesting. It appears that both strains contain different numbers of TAL effector gene clusters, and orthologs appear to be encoded at different chromosomal and/or plasmidic locations (e.g. chromosomal Tal3-XcmH1005 versus plasmidic Tal8aXcmN1003).

We thank the reviewer for this suggestion. We further characterized the genome sequences and compared genome context of two strains as presented in new Figure 1 and Supplementary Table S3. We have also added the description in the text.

-A scheme showing where the different TAL effector genes are encoded could be included in Figure 1.

We have presented this information in the new Figure 1.

-The nomenclature of TAL effector genes is confusing. Apparently, Tal3 in one strain is an ortholog of Tal8a in the other strain, in which then Tal3 corresponds to Tal4 in the first strain. This reviewer strongly suggests adapting a universal nomenclature, which will help in the future to keep track of the various TAL effector genes in different isolates. Such a nomenclature has been, for example, suggested by Grau et al. 2016 (Sci. Rep. 6: 21077).

We have updated the names with those published previously where applicable and provided in Supplementary Table S3 the corresponding AnnoTALE designations for all of the *tal* genes.

-Figure 1: It's hard if not impossible to distinguish protein-coding genes (blue) from rRNA genes (lavender), at least at the current resolution. Anyway, one may ask if these plots

provide any useful information or if they merely fulfill esthetic purposes? By the way, only chromosomes are shown in the figure, not the whole genomes.

We agree with the reviewer. We have replaced these circles with more informative genome alignments in new Figure 1A.

-Line 253: The data presented merely confirm one EBE (SWEET10D) as being required for AvrB6-induced expression but not for the other two genes.

Thanks for pointing this out. We have changed three genes to GhSWEET10D.

-Line 338: Here, it's more common to use the term "query" instead of "bait".

We have changed the term to "query" as suggested.

Reviewer 3 comments:

By comparing transcriptome data and TAL EBE predicted by computational analysis, the authors found that GhSWEET10 is one of the targets of AvrB6, which is a TAL effector in XcmH1005. Throughout the experiments described in this manuscript, it is nicely demonstrated that induction of GhSWEET10 expression by AvrB6 is required for normal BBC symptom development upon infection by Xcm. It has been suggested that sucrose transport mediated by SWEET proteins plays a role in providing the bacteria with a carbon source. Based on this, the authors determined the sugar transport activity of GhSWEET10.

We thank the reviewer's comments on our manuscript.

-Using a previously established heterologous system with the human HTK293T cells, the authors confirmed that GhSWEET10 has an ability to facilitate sucrose uptake into the cells. However, according to the hypothesis mentioned above, the role of SWEET proteins in the pathogen infection should be related to its sugar export activity. Thus, the authors need to determine whether GhSWEET10 has a sugar transport activity at least in a heterologous system such as *Xenopus* oocytes. Or an *in planta* evidence could be supported by measuring sugar concentrations in apoplast after overexpressing GhSWEET10 as described in Zhou et al. (Plant J, 2015, 82, 632-643).

We thank the reviewer for the suggestion. We have included *in planta* data by overexpressing GhSWEET10 into *N. benthamiana* and measuring the sugar concentration in its apoplast (see Fig. S6B). Indeed, overexpression of GhSWEET10 increased apoplast sugar concentration.

-In addition, it has been shown that SWEET proteins transport glucose and fructose in addition to sucrose. So I would suggest the authors to determine whether GhSWEET10 also transports these sugars.

We have tested whether GhSWEET10 could transport glucose in yeast and found that *GhSWEET10D* did not restore the growth defect of yeast EBY4000, a hexose transport-deficient

yeast strain, on the medium supplemented with 2% glucose or 2% fructose as the sole carbon source, indicating that GhSWEET10D does not mediate efficient fructose or glucose transport (Fig S6B).

-It was recently reported that Arabidopsis SWEET proteins in the clade III transport the plant hormone gibberellin (Kanno et al., Nat Commun, 2016, 7, 13245). It would be helpful to readers to cite this report in the present manuscript.

We have added this reference in the manuscript as suggested.

-Finally, the authors showed that different GhSWEET proteins in the clade III are induced depending on different Xcm field isolates. This suggests that clade III SWEETs are generally important for Xcm strains to cause BBC and that some field isolates evolved new TAL effectors to target new SWEET genes. This is an interesting hypothesis. However, I think more careful discussion (for example the presence of unidentified substrate(s) for SWEETs) would be required because it is not clear that all the GhSWEETs in the clade III have the same function (at least biochemically).

We thank the reviewer for the comment and have integrated this point into the discussion section.

REVIEWERS' COMMENTS:

Reviewer #1 (Remarks to the Author):

This manuscript is a revision of a previous version. The authors have addressed most of the many comments that the reviewers had. They changed the text in many instances and added a few additional experimental data which improved the manuscript. This work will be an important contribution to the field.

Only very minor issues remain.

It is very welcome that the authors now include the AnnoTALE nomenclature to avoid confusion with the similar naming of TALEs in different strains! Did they make sure that these names are unique (i.e. that no other TALEs will get these names)? If not, I advise to contact the curator of AnnoTALE (Jan Grau, Martin-Luther-University Halle, Germany; E-mail: grau@informatik.uni-halle.de) to coordinate the naming and reserve names for these particular TALEs.

Figure 1B: the genes are written in italics (which is fine), but some of them (PthN2, PthN', PthN) start with a capital letter in the figure, but with a small letter in the main text). Please correct the spelling in the figure.

Figure 2, Figure 3G, Figure S3, Figure S4, and Figure S9: Despite that the authors state in the rebuttal that the EBE is defined without the initial T (which is recognized by the N-terminal TALE domain and not the RVDs), all five figures include the "T" as part of the EBE. This is very confusing! The "T" is crucial for all TALEs from *Xanthomonas* with a standard N-terminal domain and a key element when predicting target sites. Even the authors' own methods section (page 21) when describing "EBE prediction" uses the "T". It would only make sense to include the "T" into the EBE designation. In contrast, in Figure 4A, 4G, and 4F, no initial "T" is indicated for the EBE, although it would make sense to show whether there is a "T" in the target sequence to assess whether the TALE would recognize it.

My suggestion to solve this is: just go ahead and re-define what makes an EBE (RVD-specificity + initial "T") in the introduction and include the "T" in all Figures. Otherwise, you need to describe that you labelled or used the "EBE plus initial T" in all instances including the methods section. The third alternative to not label the "T" in all instances would be a terrible choice, because it would blur the actual TALE-binding specificity and make it highly difficult for everyone to assess the data.

Figure 7D: according to the authors, an asterisk indicates significant (please indicate by which statistical means) induction of the GhSWEET14a gene in comparison to the water control. While this might be technically correct, it is impossible to fully assess the data, because some of the induction levels are so low that practically no induction can be seen, although they still have an asterisk (H1005Davrb6). Why not display the fold-induction levels in logarithmic scale? Furthermore, it is not likely that the gene induction via the above mentioned strain is significant (at least in the biological sense), because this strain can not cause water soaking (Fig. 3B). If the authors' argument is that GhSWEET14a is an alternative target that might support Xcm growth (which is likely to be true), then the gene induction of some of the strains (H1005 & H1005Davrb6) is not sufficient. I guess that the asterisk is only an artefact of the calculation which was used and that the induction level for these strains is background noise.

The new experiment (Figure S2) to show that CHX suppresses protein synthesis is not an appropriate control for Figure 3D, because the experimental designs are very different (*Xanthomonas* in Figure 3D and *Agrobacterium* in Figure S2). Furthermore, the control was not done in parallel to the experiment.

Reviewer #2 (Remarks to the Author):

As I stated during the first round of the reviewing process, this is a well-performed and interesting study. As far as I can see all my concerns have been properly addressed in the revised manuscript, which to me should be accepted for publication.

Reviewer #3 (Remarks to the Author):

In the revised manuscript, the authors provided new data showing that overexpression of GhSWEET10 in *Nicotiana benthamiana* resulted in increased sucrose contents in apoplast. The authors also showed that GhSWEET10 did not mediate fructose or glucose uptake when expressed in the yeast EBY4000 strain. These data support the *in vivo* function of GhSWEET10 as a sucrose exporter. The authors have appropriately responded to all my comments and I am satisfied with the present manuscript.

Responses to reviewers

Reviewer #1 comments:

This manuscript is a revision of a previous version. The authors have addressed most of the many comments that the reviewers had. They changed the text in many instances and added a few additional experimental data which improved the manuscript. This work will be an important contribution to the field. Only very minor issues remain.

We thank the reviewer's comments on our revised manuscript.

It is very welcome that the authors now include the AnnoTALE nomenclature to avoid confusion with the similar naming of TALEs in different strains! Did they make sure that these names are unique (i.e. that no other TALEs will get these names)? If not, I advise to contact the curator of AnnoTALE (Jan Grau, Martin-Luther-University Halle, Germany; E-mail: grau@informatik.uni-halle.de) to coordinate the naming and reserve names for these particular TALEs.

We are happy that the reviewer welcomes the AnnoTALEs nomenclature. We have contacted Dr. Jan Grau to coordinate and reserve the names for these TALEs.

Figure 1B: the genes are written in italics (which is fine), but some of them (PthN2, PthN', PthN) start with a capital letter in the figure, but with a small letter in the main text). Please correct the spelling in the figure.

We have corrected the spelling in the figure.

Figure 2, Figure 3G, Figure S3, Figure S4, and Figure S9: Despite that the authors state in the rebuttal that the EBE is defined without the initial T (which is recognized by the N-terminal TALE domain and not the RVDs), all five figures include the "T" as part of the EBE. This is very confusing! The "T" is crucial for all TALEs from *Xanthomonas* with a standard N-terminal domain and a key element when predicting target sites. Even the authors' own methods section (page 21) when describing "EBE prediction" uses the "T". It would only make sense to include the "T" into the EBE designation. In contrast, in Figure 4A, 4G, and 4F, no initial "T" is indicated for the EBE, although it would make sense to show whether there is a "T" in the target sequence to assess whether the TALE would recognize it. My suggestion to solve this is: just go ahead and re-define what makes an EBE (RVD-specificity + initial "T") in the introduction and include the "T" in all Figures. Otherwise, you need to describe that you labelled or used the "EBE plus initial T" in all instances including the methods section. The third alternative to not label the "T" in all instances would be a terrible choice, because it would blur the actual TALE-binding specificity and make it highly difficult for everyone to assess the data.

We have followed the reviewer's suggestion and re-defined what makes an EBE in the introduction and modified Figure 4.

Figure 7D: according to the authors, an asterisk indicates significant (please indicate by which statistical means) induction of the GhSWEET14a gene in comparison to the water control. While

this might be technically correct, it is impossible to fully assess the data, because some of the induction levels are so low that practically no induction can be seen, although they still have an asterisk (H1005Davrb6). Why not display the fold-induction levels in logarithmic scale? Furthermore, it is not likely that the gene induction via the above mentioned strain is significant (at least in the biological sense), because this strain can not cause water soaking (Fig. 3B). If the authors' argument is that GhSWEET14a is an alternative target that might support Xcm growth (which is likely to be true), then the gene induction of some of the strains (H1005 & H1005Davrb6) is not sufficient. I guess that the asterisk is only an artefact of the calculation which was used and that the induction level for these strains is background noise.

We have indicated the statistical means that were used in Figure 7D. We agree with this reviewer that it is likely that *GhSWEET14a* and *GhSWEET14b* are alternative targets for other *Xcm* isolates. Considering the relative low induction folds for *GhSWEET9a*, *GhSWEET11*, *GhSWEET12a* and *GhSWEET12b* by most of *Xcm* isolates, when the P value was set as <0.01, the induction for these genes were not statistically significant. We have noted this in the figure legends.

The new experiment (Figure S2) to show that CHX suppresses protein synthesis is not an appropriate control for Figure 3D, because the experimental designs are very different (*Xanthomonas* in Figure 3D and *Agrobacterium* in Figure S2). Furthermore, the control was not done in parallel to the experiment.

We have noted in the manuscript that this experiment was done separately to show that CHX treatment effectively blocked protein synthesis in cotton.

Reviewer 2 comments:

As I stated during the first round of the reviewing process, this is a well-performed and interesting study. As far as I can see all my concerns have been properly addressed in the revised manuscript, which to me should be accepted for publication.

We thank the reviewer for the comments.

Reviewer 3 comments:

In the revised manuscript, the authors provided new data showing that overexpression of GhSWEET10 in *Nicotiana benthamiana* resulted in increased sucrose contents in apoplast. The authors also showed that GhSWEET10 did not mediate fructose or glucose uptake when expressed in the yeast EBY4000 strain. These data support the *in vivo* function of GhSWEET10 as a sucrose exporter. The authors have appropriately responded to all my comments and I am satisfied with the present manuscript.

We thank the reviewer for the comments.